# Differentially Private Learning Needs Hidden State (Or Much Faster Convergence)

**Jiayuan Ye, Reza Shokri**
Department of Computer Science
National University of Singapore
`{jiayuan, reza}@comp.nus.edu.sg`

## Abstract

Prior work on differential privacy analysis of randomized SGD algorithms relies on composition theorems, where the implicit (unrealistic) assumption is that the internal state of the iterative algorithm is revealed to the adversary. As a result, the Rényi DP bounds derived by such composition-based analyses linearly grow with the number of training epochs. When the internal state of the algorithm is hidden, we prove a converging privacy bound for noisy stochastic gradient descent (on strongly convex smooth loss functions). We show how to take advantage of privacy amplification by sub-sampling and randomized post-processing, and prove the dynamics of privacy bound for "shuffle and partition" and "sample without replacement" stochastic mini-batch gradient descent schemes. We prove that, in these settings, our privacy bound converges exponentially fast and is substantially smaller than the composition bounds, notably after a few number of training epochs. Thus, unless the DP algorithm converges fast, our privacy analysis shows that hidden state analysis can significantly amplify differential privacy.

## 1 Introduction

Machine learning models leak sensitive information about their training data [39, 11]. To protect user privacy, the widely-used differentially private training algorithm, DP-SGD [1], adds carefully calibrated noise in each step of updating model parameters. This randomness guarantees that the models trained on any two neighboring datasets are indistinguishable in probability distributions. To quantify this indistinguishability, the DP analysis *bounds* the (moment of) likelihood ratio between a pair of models trained on any two neighboring datasets. Improving this privacy analysis is crucial for obtaining a higher (train and test) accuracy of the output model, under a constrained privacy budget.

The main-stream analysis of privacy loss in DP-SGD is based on composition theorems, which quantify the total privacy loss of the training process across all its iterations. Given that privacy-preserving learning via DP-SGD usually suffers from a slow convergence in empirical prediction accuracy [10], the final bound could be significantly large (and loose). This analysis worsens the privacy-accuracy trade-off by overestimating the magnitude of required noise for DP training.

To alleviate this problem, one popular direction is to design new variants of the DP-SGD algorithm, that converge faster, i.e. require a smaller number of iterations to reach a stable training accuracy. This includes works that derive privacy preserving variants of fast optimization algorithms, such as performing DP-SGD with momentum [25], adaptive gradient clipping [3, 30], adaptive selection of step-size and noise scale [4], and using pre-trained or hand-crafted features [34]. All these approaches aim for a *faster training convergence* for the DP learning algorithm, such that the composition analysis is applied to a smaller number of iterations and the total bound remains small.

36th Conference on Neural Information Processing Systems (NeurIPS 2022).

Another line of work focuses on directly improving the privacy analysis of the DP-SGD algorithm. To this end, prior work [18, 5, 15] suggest that hiding the internal state of noisy (S)GD, therefore analyzing the privacy bound for releasing only the last iterate results in a more accurate estimation of privacy loss. By applying the last-iterate privacy analysis [18], Feldman et al. [19] design new differentially private algorithms that achieve theoretically optimal excess risk with linear runtime for convex optimization, which is better than the prior privacy utility trade-off for the noisy SGD algorithm [7]. However, the analysis applies only to one *single* epoch of training, thus it is unclear how such training performs in practical learning settings (which require multiple epochs of training). Chourasia et al. [15] derive a strong privacy guarantee for hidden-state noisy GD over many training epochs, under strongly convex loss function. However, the analysis is limited to computationally expensive GD [23], and extension of it to stochastic mini-batch training would fail to model the privacy amplification by post-processing [18, 5] and mini-batch sub-sampling [12, 27, 7, 1, 5].

Thus, under the hidden-state assumption over many epochs of training, the challenge is to compute a small privacy bound for differentially private stochastic gradient descent algorithms by taking advantage of privacy amplifications due to sub-sampling and randomized updates over mini-batches.

***Contributions***. In this paper, we tackle this challenge, and bridge the gaps in the prior work on hidden-state (last-iterate) privacy analysis. We model the privacy dynamics of noisy SGD, and show the privacy amplification for private (hidden state) machine learning due to stochastic mini-batch selection. **(i)** As we also show in this figure, for multi-epoch noisy *stochastic mini-batch* gradient descent under *"shuffle and partition"* and iterative *"sampling without replacement"*, we prove new converging privacy bounds (for strongly convex smooth loss function) that significantly improves over the prior bounds [5, 29, 15]. Our proof relies on our new bounds for the privacy amplification by post-

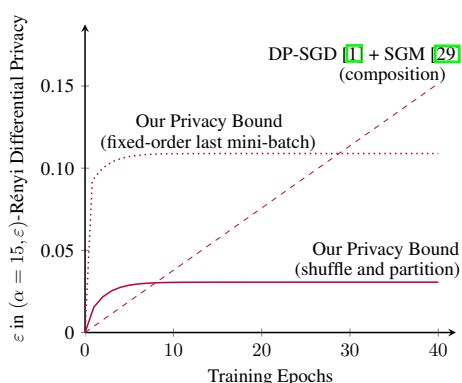

processing (Lemma 3.2) and sub-sampling (Theorems 4.2 and 4.3). **(ii)** For the special case of full gradient descent, our new approach (via proving better bounds for the privacy amplification by randomized post-processing) results in a strictly tighter bound than the prior work [15] with a different proof (Appendix D.7). The key insight of our new proof is that we can break down one noisy GD update into two consecutive steps: a noisy GD update with smaller noise scale, followed by pure additive Gaussian noise (randomized post-processing).

Our results show that, to obtain a tighter privacy bound, it is crucial to apply our hidden-state privacy dynamics analysis for learning tasks with slow to moderate convergence. Alternatively, it is not so costly to use composition-based privacy analysis when the training process converges very quickly.

## 2 An overview of the problem and our approach

We analyze differential privacy loss of the noisy (stochastic) mini-batch gradient descent algorithm, when only the last iterate parameters $\theta_K^0$ are visible (see Algorithm 1).[1] We consider two mini-batch generation variants: **(1)** "shuffle and partition" (widely implemented in privacy libraries [40, 24]); **(2)** "sample without replacement" (analyzed extensively in prior results [12, 27, 5, 37] despite its computational cost). Our ultimate goal is to prove a *worst-case* upper bound of the Rényi divergence between distributions of *last-iterate* parameters $\theta_K^0$ and $\theta'^0_K$ trained on *any* two neighboring datasets.

The prior work Chourasia et al. [15] has shown a tight converging hidden-state privacy dynamics analysis for full-batch noisy gradient descent. However, the sub-sampling steps that are unique to the stochastic mini-batch gradient descent, and their privacy benefits, are not modeled nor quantified in such bounds and the follow-up work. Would a naive extension of the prior GD analysis to the stochastic mini-batch setting result in a tight bound? To this end, we can view the updates that *involve* the (sensitive) differing data record (between neighboring datasets) as gradient descent on a *smaller*, iteratively *changing* dataset of size $b$ (i.e., the size of a mini-batch). This extension results in the following naive privacy baseline.

---

[1]See Appendix B for the preliminaries about differential privacy and necessary tools for our analysis.

---

**Algorithm 1** $\mathcal{A}_{\text{Noisy-mBGD}}$: Noisy (Stochastic) mini-batch Gradient Descent

---

1: **Input:** Dataset $D = (\mathbf{x}_1, \mathbf{x}_2, \cdots, \mathbf{x}_n)$. Parameter space $\theta \in \mathbb{R}^d$. Loss function $\ell(\theta; \mathbf{x})$. Stepsize $\eta$. Noise standard deviation $\sigma$. mini-batch size $b$. Initial parameters $\theta_0^0$ sampled from an arbitrary distribution $p_0(\theta)$.
2: **Batch Generation:** obtain mini-batches $B_k^j$ of size $b$, for $j = 0, \cdots, n/b-1$; for epochs $k = 0, \cdots, K-1$.
3:    ▷ **If "Shuffle and partition"**: randomly partition $n$ data indices into $B^0, \cdots, B^{n/b-1}$, let $B_k^j = B^j$.[3]
4:    ▷ **If Sample without replacement**: resample $b$ different indices from $\{1, \cdots, n\}$ to obtain every $B_k^j$.
5: **for** epoch $k = 0, 1, \cdots, K-1$ **do**
6:    **for** iteration $j = 0, 1, \cdots, n/b-1$ **do**
7:       $\theta_k^{j+1} = \theta_k^j - \eta \cdot g\left(\theta_k^j; B_k^j\right) + \sqrt{2\eta\sigma^2} \cdot \mathcal{N}\left(0, \mathbb{I}_d\right)$    where    $g\left(\theta_k^j; B_k^j\right) = \frac{1}{b} \sum_{i \in B_k^j} \nabla\ell(\theta_k^j; \mathbf{x}_i)$
8:    $\theta_{k+1}^0 = \theta_k^{n/b}$
9: **Output:** $\theta_K^0$

---

**Theorem 2.1** (Naive Extension of Chourasia et al. [15] bound to SGD). *If the loss function $\ell(\theta; \mathbf{x})$ is $\lambda$-strongly convex and $\beta$-smooth, and its gradient has $\ell_2$-sensitivity $S_g$, then Algorithm 1 under "shuffle and partition" and step-size $\eta < \frac{2}{\lambda+\beta}$ satisfies $(\alpha, \varepsilon)$-Rényi DP with $\varepsilon \leq \frac{\alpha S_g^2}{\lambda \sigma^2 b^2}(1 - e^{-\lambda\eta K/2})$.*

However, as we also show in Figure 1, this naive privacy dynamics baseline *slowly* converges to a *huge* constant, which is significantly worse than the bounds derived by composition [29, 1].[2] This is because this naive baseline *fails to* capture the privacy amplification due to the stochasticity of mini-batch sub-sampling, and the iterative noisy updates which amplify the privacy of preceding mini-batches (referred to as amplification by post-processing). In this paper, we show how to compute a much tighter differential privacy bound for noisy SGD, under the hidden-state assumption, while taking advantage of these privacy amplifications. Our methodology is novel and quantifies the hidden-state privacy amplification due to iterative data (re)sampling throughout the training process (over multiple epochs of training). This solves the limitation of the prior work on privacy amplification (by iteration) which focuses on a single epoch, and uses composition theorems across epochs (thus, not modeling the privacy of hidden-state iterative resampling). Finally, our methodology is generic and applies to the special case of full-batch noisy gradient descent, which enables a strictly tighter privacy bound than the prior work [15] under the same assumptions (Appendix D.7).

Our approach is as follows. We decompose the distribution of last iterate parameters $\theta_K^0$ in Algorithm 1 as a mixture of conditional distributions $p(\theta_K^0) = \sum_B p(B) \cdot p(\theta_K^0|B)$, given any possible mini-batch sequence $B$. We first analyze the Rényi privacy loss for a *fixed mini-batch sequence* while modeling privacy amplification by randomized post-processing (i.e., how much the randomized gradient update improve the privacy due to updates on preceding mini-batches). We then quantify the privacy amplification by subsampling under *stochastic mini-batches*.

***Privacy amplification by randomized post-processing.*** To start, we consider Algorithm 1 without the effect of stochasticity in the mini-batch sampling process. That is, we first assume that the mini-batch sequence used in the algorithm is fixed (by an arbitrary order). When a mini-batch contains only the indices of shared data records between two neighboring datasets, then the deterministic mini-batch gradient descent mapping does not cause any additional privacy loss. Addition of the Gaussian noise that follows this deterministic update, however, serves as a randomized post-processing which *decreases* the Rényi divergence between the two processes (associated to neighboring datasets). To compute this privacy amplification, we precisely model the change of parameter distributions with the Fokker-Planck equation (for diffusion process with zero drift).

***Hidden-state privacy amplification by sub-sampling.*** Although privacy amplification by a *single* sub-sampling operation is well-studied [12, 27, 7, 1, 5, 37, 29, 20, 21], to the best of our knowledge, no prior bounds are applicable to multiple epochs of *hidden-state* iterative resampling. This is because

---

[2] A recent concurrent work [32] also follows this approach to extend the GD analysis [15] to SGLD setting, and proves a similar bound to our naive privacy dynamics baseline Theorem 2.1. However, there is a slight difference between [32, Corollary 3.3] and our Theorem 2.1, due to the flawed assumption in [32, Lemma 3.4] that the LSI constant proved in Chourasia et al. [15] (that *only* holds for GD process) also holds for SGLD process (that takes the form of a more complicated mixture distribution). See more details in Appendix C.

[3] For simplicity of presentation, we assume $b$ divides $n$. If $n/b$ is not an integer, and if the algorithm ignores the last $n - \lfloor n/b \rfloor \cdot b$ data points, then our privacy dynamics bound holds by replacing $n/b$ with $\lfloor n/b \rfloor$.

under the hidden-state assumption, the number of mixing components in the last-iterate parameter distribution grows exponentially with the number of epochs $K$, i.e., [number of possible values for one mini-batch]$^{K \cdot n/b}$, which makes the mixture distribution very difficult to analyze. In this paper, to study the exponentially many mixture components, we derive recursions for the divergence between mixture distributions after one epoch. We use the joint convexity of exponentiated Rényi divergence to bound how much smaller the Rényi divergence between mixture distributions (for model parameters at one epoch) is compare to the worst case Rényi divergence across any pair of their mixture components (representing the preceding epoch). This recursion quantifies the hidden-state privacy amplification by sub-sampling, and enables a significantly smaller Rényi DP bound for noisy stochastic mini-batch gradient descent, than the composition of sub-sampled mechanisms over multiple epochs (Figure 2).

## 3 Privacy dynamics for fixed-ordering noisy mini-batch gradient descent

In this section, we quantify the privacy amplification by randomized post-processing in Algorithm 1, during iterations that do not access the sensitive differing data between neighboring datasets. We then combine it with RDP composition for the remaining iterations and prove a converging privacy bound.

### 3.1 Privacy amplification by randomized post-processing (additive Gaussian noise)

We first explain the key lemma that proves exponentially decaying Rényi privacy loss under additive Gaussian noise post-processing, when the parameter distributions satisfy the log-Sobolev inequality. This is a new bound for the well-known "privacy amplification by iteration" phenomenon [18, 5].

**Lemma 3.1.** *Let $\mu, \nu$ be two distributions on $\mathbb{R}^d$. Let $f : \mathbb{R}^d \to \mathbb{R}^d$ be a measurable mapping on $\mathbb{R}^d$. We denote $\mathcal{N}(0, 2t\sigma^2 \cdot \mathbb{I}_d)$ to be the standard Gaussian distribution on $\mathbb{R}^d$ with covariance matrix $2t\sigma^2 \cdot \mathbb{I}_d$. We denote $p_t(\theta)$ and $p_t'(\theta)$ to be the probability density functions for the distributions $f_\#(\mu) * \mathcal{N}(0, 2t\sigma^2 \mathbb{I}_d)$ and $f_\#(\nu) * \mathcal{N}(0, 2t\sigma^2 \mathbb{I}_d)$ respectively, where $f_\#(\mu), f_\#(\nu)$ denote the push forward distributions of $\mu, \nu$ under mapping $f$. Then if $\mu$ and $\nu$ satisfy log-Sobolev inequality with constant $c$, and if the mapping $f$ is $L$-Lipschitz, then for any order $\alpha > 1$,*

$$\frac{\partial}{\partial t} R_\alpha \left( p_t(\theta) \| p_t'(\theta) \right) \leq -c_t \cdot 2\sigma^2 \cdot \left( \frac{R_\alpha(p_t(\theta) \| p_t'(\theta))}{\alpha} + (\alpha - 1) \cdot \frac{\partial}{\partial \alpha} R_\alpha(p_t(\theta) \| p_t'(\theta)) \right), \quad (1)$$

*where $c_t = \left( \frac{L^2}{c} + 2t\sigma^2 \right)^{-1}$ is the log-Sobolev inequality constant for distributions $p_t(\theta)$ and $p_t'(\theta)$.*

*Proof Sketch.* The proof starts by modelling Gaussian noise as a diffusion process with *zero* drift. We then bound the rate of Rényi divergence with the LSI constant for the process (following prior works [35, 15]). However, instead of assuming a fixed LSI constant $c$ for all intermediate distributions, we prove a more precise LSI constant $c_t$ that changes with $t$. Complete proof is in Appendix D.1. □

The partial differential inequality in Lemma 3.1 quantifies the amplification of Rényi privacy loss under additive Gaussian noise. By solving Equation (1) on $t \in [0, \eta]$, we prove the following lemma that models recursive privacy dynamics during one step of noisy mini-batch gradient descent.

**Lemma 3.2.** *Let $D, D'$ be an arbitrary pair of neighboring datasets that differ in the $i_0$-th data point (i.e. $x_{i_0} \neq x_{i_0}'$). Let $B_k^j$ be a fixed mini-batch used (in iteration $j$ of epoch $k$) in Algorithm 1, which contains $b$ indices sampled from $\{1, \cdots, n\}$. We denote $\theta_k^j$ and $\theta_k'^j$ as the intermediate parameters in Algorithm 1 on input datasets $D$ and $D'$, respectively. If the distributions of $\theta_k^j$ and $\theta_k'^j$ satisfy log-Sobolev inequality with a constant $c$, and if the mini-batch GD mapping $f(\theta) = \theta - \eta \cdot \frac{1}{b} \cdot \sum_{i \in B_k^j} \ell(\theta; \mathbf{x}_i)$ is $L$-Lipschitz for parameters $\theta$, then the following recursive bound for Rényi divergence holds.*

$$\frac{R_\alpha(\theta_k^{j+1} \| \theta_k'^{j+1})}{\alpha} \leq \begin{cases} \frac{R_{\alpha'}(\theta_k^j \| \theta_k'^j)}{\alpha'} \cdot \left( 1 + \frac{c \cdot 2\eta\sigma^2}{L^2} \right)^{-1} & \text{if } i_0 \notin B_k^j \\ \frac{R_\alpha(\theta_k^j \| \theta_k'^j)}{\alpha} + \frac{\eta S_g^2}{4\sigma^2 b^2} & \text{if } i_0 \in B_k^j \end{cases} \text{ with } \alpha' = \frac{\alpha - 1}{1 + \frac{c \cdot 2\eta\sigma^2}{L^2}} + 1. \quad (2)$$

*Proof Sketch.* When $i_0 \notin B_k^j$, we apply Lemma 3.1 on the deterministic mini-batch gradient descent update $f$, and solve Equation (1) to obtain recursive Rényi privacy bound. This proof is similar to

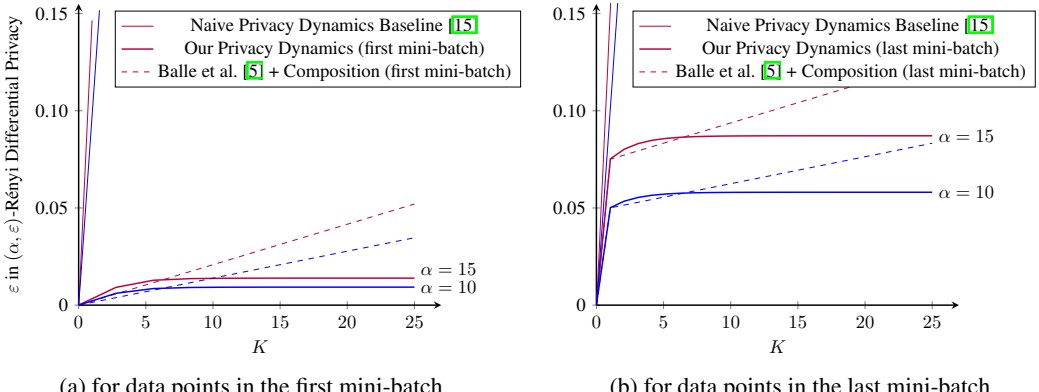

(a) for data points in the first mini-batch        (b) for data points in the last mini-batch

Figure 1: Rényi privacy loss of fixed-ordering noisy mini-batch gradient descent over $K$ epochs, which repeatedly goes over a fixed sequence of mini-batch partition in each epoch. We show the privacy loss for data points in the first mini-batch $B^0$ and the last mini-batch $B^{n/b-1}$ of each pass. We show the $\varepsilon$ in the $(q, \varepsilon)$-RDP guarantee derived by our privacy dynamics analysis (bold lines), the naive privacy dynamics baseline (thin lines), and the privacy amplification by mixing and diffusion analysis [5] combined with composition (dashed line). We evaluate under the following setting: RDP order $\alpha \in \{10, 15\}$; $\lambda$-strongly convex loss function with $\lambda = 1$; $\beta$-smooth loss function with $\beta = 4$; gradient sensitivity $S_g = 4$; size of the data set $n = 50$; step-size $\eta = 0.02$; noise variance $\sigma^2 = 4$, mini-batch size $b = 2$. We use Theorem 3.3 for our privacy dynamics; Theorem 2.1 for naive privacy dynamics baseline; and [5, Theorem 5] for Balle et al. [5] + Composition (details are in Appendix D.6).

that of [15, Theorem 2], however, dealing with the new LSI constant $c_t$ that changes with $t$ introduces an additional technical difficulty. When $i_0 \in B_k^j$, we use composition theorem to prove the additive recursion in the second row of Equation (2). The complete proof is in Appendix D.2. □

***Comparison with prior amplification by post-processing bound.*** Prior to this work, for convex and smooth loss functions, Feldman et al. [18] derive a tight bound for the amplification by post-processing in noisy stochastic gradient descent. In Appendix D.4, we show that our new recursive amplification bound could recover this known *tight* bound for convex smooth loss functions, while using a different proof. For (a more restrictive setting of) *one* epoch of noisy SGD on *strongly convex* loss function, Balle et al. [5] further improve the bound in Feldman et al. [18] via a careful coupling-based approach. However, we do not see an easy way to extend the bound in Balle et al. [5] to *multiple* epochs except by using Rényi DP composition (which would give a linearly worsening Rényi DP bound with the number of epochs). By contrast, our recursive amplification bound easily applies to multiple epochs (under hidden state assumption, and without requiring composition over the epochs), and enables converging privacy dynamics for Algorithm 1 on strongly convex smooth loss functions (as discussed in Section 3.2).

## 3.2 Improved privacy dynamics for fixed-ordering noisy mini-batch gradient descent

By using the recursive privacy bound Lemma 3.2 and the non-overlapping property of mini-batch partitions of the dataset, we prove position-dependent Rényi DP bounds for Algorithm 1 as follows.

**Theorem 3.3** (Privacy dynamics under strongly convex smooth loss). *Conditioned on a fixed sequence of partitioned mini-batches $B^0, \cdots, B^{n/b-1}$ in Line 3, if the loss function is $\lambda$-strongly convex, $\beta$-smooth and its gradient has $\ell_2$-sensitivity $S_g$, then running Algorithm 1 for $K \geq 1$ epochs with step-size $\eta < \frac{2}{\lambda+\beta}$, satisfies $(\alpha, \varepsilon)$-Rényi DP for data points in the batch $B^{j_0}$, with*

$$\varepsilon \leq \varepsilon_0^{\lfloor \frac{n}{2b} \rfloor}(\alpha) \cdot \frac{1 - (1 - \eta\lambda)^{2 \cdot (K-1) \cdot (n/b - \lfloor \frac{n}{2b} \rfloor)}}{1 - (1 - \eta\lambda)^{2 \cdot (n/b - \lfloor \frac{n}{2b} \rfloor)}} + \varepsilon_0^{n/b-j_0}(\alpha) \tag{3}$$

*where $\varepsilon_0^j(\alpha) = \frac{\alpha \eta S_g^2}{4\sigma^2 b^2} \cdot (1 - \eta\lambda)^{2 \cdot (j-1)} \cdot \frac{1}{\sum_{s=0}^{j-1} (1-\eta\lambda)^{2s}}$ for any $j = 1, \cdots, \frac{n}{b}$ (we assume $\frac{n}{b} \geq 2$).*

*Proof Sketch.* We first prove that the distribution of parameters $\theta_k^j$ satisfies LSI with a constant $c_k^j$ that depends on $k, j$. Then we plug the LSI constants into Lemma 3.2 and prove a recursive privacy bound

for data points in each batch $B^{j_0}$. Finally, by carefully choosing which recursion to use and solve, we obtain the privacy bound in the theorem statement. The complete proof is in Appendix D.5. □

The above new privacy bound Theorem 3.3 quantifies the privacy amplification during iterations that only access the shared data points between neighboring datasets, and it is significantly smaller than the naive privacy dynamics baseline derived from Chourasia et al. [15] (that does not capture this additional privacy amplification). In Figure 1, we observe that for data points in the first batch $B^0$, our new privacy dynamics bound is smaller by a multiplicative factor of approximately $n/b$ (where $b/n$ is the batch sampling ratio), and the improvement for data points in the last-batch is also significant.

***Bound improvement compared with Balle et al. [5].*** In the first epoch (when $K = 1$), our privacy dynamics bound (based on recursive scheme) in Theorem 3.3 is of same order as the privacy amplification bound Balle et al. [5, Theorem 5], as shown in Figure 1. However, for training with multiple epochs, Balle et al. [5] use a coupling-based approach, and we do not see an easy way to extend their analysis except by using Rényi DP composition [28] over the epochs, which gives a linearly accumulating Rényi DP bound for multiple epochs (as $K$ increases). On the contrary, as shown in Figure 1, our improved privacy dynamics bound converges to a constant, thus significantly improves over the Rényi DP composition of Balle et al. [5, Theorem 5].

# 4 Privacy dynamics for noisy stochastic mini-batch gradient descent

In this section, we further improve over the position-dependent privacy dynamics analysis in Section 3, by incorporating the effect of amplification of privacy loss due to the stochastic mini-batch sampling. We first prove the following lemma by the convexity of $f$-divergence with $f(x) = x^\alpha$.

**Lemma 4.1** (Joint convexity of scaled exponentiation of Rényi divergence). *Let $\mu_1, \cdots, \mu_m$ and $\nu_1, \cdots, \nu_m$ be distributions over $R^d$. Then for any RDP order $\alpha \geq 1$, and any coefficients $p_1, \cdots, p_m \geq 0$ that satisfy $p_1 + \cdots + p_m = 1$, the following inequality holds.*

$$e^{(\alpha-1) \cdot R_\alpha(\sum_{j=1}^m p_j \mu_j \| \sum_{j=1}^m p_j \nu_j)} \leq \sum_{j=1}^m p_j \cdot e^{(\alpha-1) \cdot R_\alpha(\mu_j \| \nu_j)} \tag{4}$$

We provide a detailed proof for Lemma 4.1 in Appendix E. This Lemma is our main tool for quantifying the additional privacy amplification under stochastic mini-batches in the rest of the section.

## 4.1 Privacy dynamics under shuffle and partition

The "shuffle and partition" batch generation scheme is commonly used for practical DP-SGD implementations in privacy libraries [24, 40]. By using Theorem 3.3 and new bounds for the privacy amplification by shuffling in Algorithm 1, we prove the following Rényi DP guarantee.

**Theorem 4.2** (Privacy dynamics under "shuffle and partition"). *If the loss function $\ell(\theta; x)$ is $\lambda$-strongly convex, $\beta$-smooth, and if its gradient has finite $\ell_2$-sensitivity $S_g$, then for $\frac{n}{b} \geq 2$, running Algorithm 1 for $K \geq 1$ epochs with step-size $\eta < \frac{2}{\lambda+\beta}$, under "shuffle and partition" batch generation scheme, satisfies $(\alpha, \varepsilon)$-Rényi DP with*

$$\varepsilon \leq \varepsilon_0^{\lfloor \frac{n}{2b} \rfloor}(\alpha) \cdot \frac{1 - (1-\eta\lambda)^{2 \cdot (K-1) \cdot (n/b - \lfloor \frac{n}{2b} \rfloor)}}{1 - (1-\eta\lambda)^{2 \cdot (n/b - \lfloor \frac{n}{2b} \rfloor)}} + \frac{1}{\alpha - 1} \cdot \log \left( \underset{0 \leq j_0 < n/b}{Avg} \; e^{(\alpha-1)\varepsilon_0^{n/b - j_0}(\alpha)} \right) \tag{5}$$

*where $\varepsilon_0^j(\alpha) = \frac{\alpha \eta S_g^2}{4\sigma^2 b^2} \cdot (1-\eta\lambda)^{2 \cdot (j-1)} \cdot \frac{1}{\sum_{s=0}^{j-1}(1-\eta\lambda)^{2s}}$ for any $j = 1, \cdots, \frac{n}{b}$ (we assume $\frac{n}{b} \geq 2$).*

The above Rényi DP bound Theorem 4.2 is always smaller than the Rényi DP bound in Theorem 3.3 for the worst-case fixed mini-batch sequence (i.e., when the differing data point is in the last mini-batch of each pass). Therefore, Theorem 4.2 quantifies the privacy amplification due to shuffling, when compared to the worst-case Rényi DP bound among all possible mini-batch sequences. In Figure 2 (a), we illustrate this amplification in more details.

---

[4]When $\alpha - 1$ and $\varepsilon_0^{n/b - j_0}(\alpha)$ are large, the second term in Equation (5) might overflow. In the experiments, we compute this log-sum-exp term with the shifted approximation $\varepsilon_0^1 + \log \underset{0 \leq j_0 < n/b}{Avg} \; e^{(\alpha-1)\left(\varepsilon_0^{n/b - j_0}(\alpha) - \varepsilon_0^1(\alpha)\right)}$.

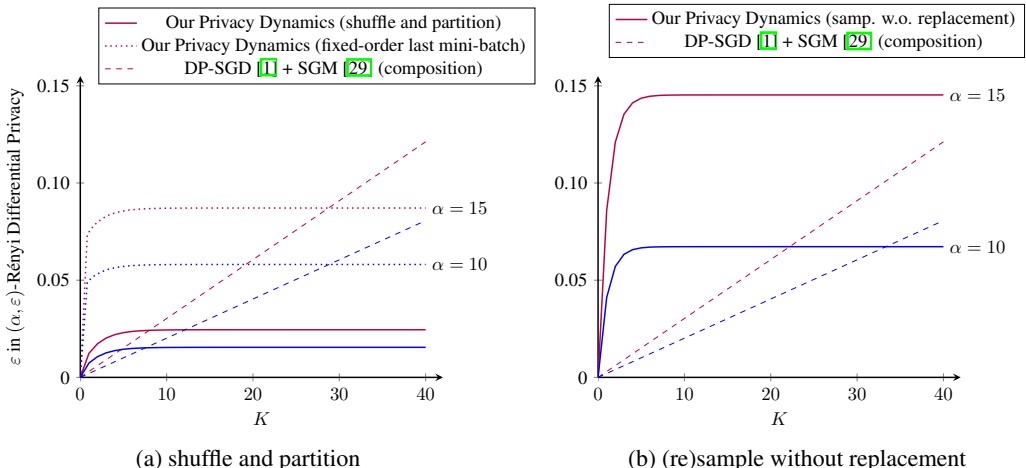

(a) shuffle and partition

(b) (re)sample without replacement

Figure 2: Rényi privacy loss of noisy (stochastic) mini-batch gradient descent over $K$ epochs. We show $\varepsilon$ in the $(q, \varepsilon)$-RDP guarantee derived by our privacy dynamics bound under "shuffle and partition" (bold lines, left plot), our privacy dynamics bound under sampling without replacement (bold lines, right plot), our privacy dynamics bound for data points in the last batch $B^{n/b-1}$ (thin dashed lines), and the baseline composition-based bound for DP-SGD [1, 29] (thin lines). We evaluate under the following setting: RDP order $\alpha \in \{10, 15\}$; $\lambda$-strongly convex loss function with $\lambda = 1$; $\beta$-smooth loss function with $\beta = 4$; finite total gradient sensitivity $S_g = 4$; size of the data set $n = 50$; step-size $\eta = 0.02$; noise variance $\sigma^2 = 4$, batch size $b = 2$. The expressions for computing the privacy bounds are: Privacy Dynamics (shuffle and partition): Theorem 4.2 [4]; Privacy Dynamics (samp. w.o. replacement): Theorem 4.3; Privacy Dynamics (last batch): Theorem 3.3 under $j_0 = n/b - 1$; Composition: derived from Section 3.3. of Mironov et al. [29] that approximately equals $\frac{b}{n} \cdot \frac{\alpha \eta S_g^2}{4\sigma^2 b^2} \cdot K$.

***Comparison with prior amplification by sub-sampling bounds [29, 20, 21].*** We now investigate how our Rényi DP bound Theorem 4.2 compares with prior bounds for privacy amplification by sub-sampled mechanisms, in terms of its amplification rate (i.e. the ratio between RDP bound for noisy mini-batch GD on sub-sampled mini-batch with size $b$, and RDP bound for full-batch noisy GD on a small dataset with size $b$). Observe that only the first term of the Rényi DP bound Equation (5) increases with the number of epochs $K$, and its growth rate is strictly smaller than $\varepsilon_0^{\lfloor \frac{n}{b} \rfloor} < \frac{b}{n} \frac{\alpha \eta S_g^2}{4\sigma^2 b^2}$. Here, the term $\frac{\alpha \eta S_g^2}{4\sigma^2 b^2}$ is exactly the Rényi DP bound for a *full-batch* noisy GD update on dataset with small size $b$. Therefore, Theorem 4.2 achieves an amplification rate of $O(\frac{b}{n})$ for *one epoch* of noisy mini-batch gradient descent under "shuffle and partition". This $O(\frac{b}{n})$ amplification rate for one epoch matches the $O(\frac{b^2}{n^2})$ amplification rate in prior bounds for one epoch of sub-sampled mechanisms [29, 20, 21], because one epoch consists of $\frac{n}{b}$ iterations.

However, prior bounds for privacy amplification by sub-sampling (such as for sub-sampled Gaussian mechanism [29] and for noisy SGD under "shuffle and partition" [20, 21]) only study a single update or a single epoch, and then rely on composition theorem for computing amplified privacy bounds for multiple epochs. Consequently, the Rényi DP bounds derived by such analyses linearly grow with the number of epochs $K$. On the contrary, our Rényi DP bound in Theorem 4.2 applies to multiple epochs with hidden intermediate state, and thus enables a converging Rényi DP bound that never exceeds a maximum value. Hence, our Rényi DP bound in Theorem 4.2 is strictly smaller than composition-based privacy bound for DP-SGD [1, 29] after $\frac{1}{\lambda\eta} + \frac{n}{b}$ epochs. This is because the first term in Equation (5) is strictly smaller than the composition-based privacy bound after $\frac{1}{\lambda\eta}$ epochs, and the second term in Equation (5) is strictly smaller than the composition-based privacy bound after $n/b$ epochs (as we explain with more details in Appendix E.1).

## 4.2 Privacy dynamics under (re)sampling mini-batch of fixed size without replacement

We now similarly analyze privacy dynamics for another variant of noisy mini-batch gradient descent (Algorithm 1), where in each iteration of each epoch, we freshly (re)sample $b$ indices without replacement from $\{1, \cdots, n\}$ to obtain a fixed-size mini-batch. That is, the mini-batches used in

different iterations (of one epoch) may overlap. This "sampling a mini-batch of fixed size without replacement" scheme is widely studied in the amplification by sub-sampling literature [5, 37, 26], as an attractive alternative that ensures fixed mini-batch size (when compared to Poisson sub-sampling).

**Theorem 4.3** (Recursive amplification by sampling without replacement). *If the loss function $\ell(\theta; x)$ is $\lambda$-strongly convex, $\beta$-smooth, and if its gradient has finite $\ell_2$-sensitivity $S_g$, then Algorithm 1 under sampling without replacement and stepsize $\eta < \frac{2}{\lambda+\beta}$ satisfies $(\alpha, \varepsilon)$-Rényi DP guarantee with*

$$\varepsilon \le \frac{1}{\alpha - 1} \log \left( S_K^0(\alpha) \right) \tag{6}$$

*where the terms $S_k^j(\alpha)$ for $k = 0, \cdots, K-1$ and $j = 0, \cdots, n/b - 1$ are recursively computed by*

$S_0^0(\alpha) = 1$; $S_k^{j+1}(\alpha) = \frac{b}{n} \cdot e^{\frac{(\alpha-1)\alpha\eta S_g^2}{4\sigma^2 b^2}} \cdot S_k^j(\alpha) + (1 - \frac{b}{n}) \cdot S_k^j(\alpha)^{(1-\eta\lambda)^2}$; *and* $S_{k+1}^0(\alpha) = S_k^{n/b}(\alpha)$.

Observe that in the above recursion, the term $S_k^{j+1}(\alpha)$ is strictly smaller than $e^{\frac{(\alpha-1)\alpha\eta S_g^2}{4\sigma^2 b^2}} \cdot S_k^j(\alpha)$ (i.e., the Rényi DP bound for one step of noisy gradient descent on dataset with size $b$), thus quantifying the privacy amplification by sampling a mini-batch of fixed size without replacement. We illustrate this privacy amplification in Figure 2 (b). However, our privacy bound under "sample without replacement" in Figure 2 (b) is larger than our privacy bound under "shuffle and partition" in Figure 2 (a), which suggests room for future improvement of the amplification rate bound. Indeed, for the special case of a single update, prior privacy amplification by subsampling bounds [37, 26] achieve better amplification rates of $O(\frac{b^2}{n^2})$.

However, prior bounds for privacy amplification by sampling generally *only* apply to *a single sub-sampled update*. Therefore, for analyzing multiple iterations of updates, prior works still rely on composition theorems, which results in linear growth of Rényi DP bound with regard to number of epochs. On the contrary, our privacy amplification bound Theorem 4.3 applies to multiple iteration of hidden-state (re)sampling without replacement steps, which enables a converging Rényi DP bound even for training an infinite number of epochs. In Figure 2 (b), we compare our Rényi DP bound Theorem 4.3 with the baseline composition-based privacy bound for DP-SGD [1, 28]. We observe that for a range of RDP orders $\alpha = 10, 15, 20$, our privacy dynamics bound significantly improves over the baseline composition-based bound (after 50 epochs).

## 5 Example: privacy dynamics for DP-SGD on regularized logistic regression

In this section, we explain a practical setting that our improved privacy dynamics analysis is applicable: training regularized logistic regression with the DP-SGD algorithm [1] (under "shuffle and partition" mini-batch sampling scheme). Note that DP-SGD algorithm under "shuffle and partition" is equivalent to our analyzed Algorithm 1 after change of notations, as discussed in Appendix F.1. For completeness, in Algorithm 2, we also provide the pseudocode for an equivalent of DP-SGD algorithm under notations in our paper.

### 5.1 How to ensure strong convexity, smoothness and finite sensitivity

Suppose that we want to train models for image classification tasks. The training dataset that we take as input is $D = (\mathbf{z}_1, \cdots, \mathbf{z}_n)$, where each data record $\mathbf{z}_i = (\mathbf{x}_i, \mathbf{y}_i)$ consists of the $d$-dimensional data input feature vector $\mathbf{x}_i \in \mathbb{R}^d$, and the label vector $\mathbf{y}_i \in \{0, 1\}^c$ (in one-hot encoding). To satisfy the necessary conditions for our privacy dynamics bound, we use regularized logistic regression to ensure strong convexity, use feature clipping to ensure smoothness, and use gradient clipping to enforce finite gradient sensitivity, as follows.

***Regularzied Logistic regression (for strong convexity).*** The loss function for regularized logistic regression in the multi-class setting (with per-class bias) is as follows.

$$\ell_\lambda(\theta; \mathbf{x}, \mathbf{y}) = \ell_0(\theta; \mathbf{x}, \mathbf{y}) + \frac{\lambda}{2} \|\theta\|_2^2 \tag{7}$$

where $\ell_0(\theta; \mathbf{x}, \mathbf{y})$ is the following logistic regression loss function.

$$\ell_0(\theta; \mathbf{x}, \mathbf{y}) = -\mathbf{y}^1 \log \left( \frac{e^{\bar{x}^T \cdot \theta_1}}{e^{\bar{x}^T \cdot \theta_1} + \cdots + e^{\bar{x}^T \cdot \theta_c}} \right) - \cdots - \mathbf{y}^c \log \left( \frac{e^{\bar{x}^T \cdot \theta_c}}{e^{\bar{x}^T \cdot \theta_1} + \cdots + e^{\bar{x}^T \cdot \theta_c}} \right) \tag{8}$$

where $\bar{\mathbf{x}} = (\mathbf{x}, 1) \in \mathbb{R}^{d+1}$ denotes the concatenation of the data feature vector $\mathbf{x}$ and 1, and $\mathbf{y} = (\mathbf{y}^1, \cdots, \mathbf{y}^c)$ is the label vector. The parameter vector is $\theta = (\theta_1, \cdots, \theta_c) \in \mathbb{R}^{(d+1)\cdot c}$ that represents the weight and the per-class bias of the linear model. The logistic regression loss function is convex, and therefore the regularized logistic regression loss function is $\lambda$-strongly convex.

***Feature Clipping (for bounding the smoothness constant).*** To ensure that the condition of loss function smoothness with regard to parameters $\theta$ is satisfied, we follow Feldman et al. [18] and normalize the data feature vector in $\ell_2$ norm, such that $\|\mathbf{x}\|_2 \leq L$. Under this data feature clipping, we prove that the logistic regression loss function (8) is $(\frac{L^2+1}{2})$-smooth in the following Proposition 5.1.

**Proposition 5.1.** *If the data feature vector $\mathbf{x}$ has bounded $\ell_2$ norm, such that $\|\mathbf{x}\|_2 \leq L$, then the unregularized logistic regression loss function $\ell_0(\theta; \mathbf{x}, \mathbf{y})$ Equation (8) is convex , $L$-Lipschitz and $\beta$-smooth with regard to parameters $\theta$, for*

$$L = \sqrt{2(L^2+1)} \tag{9}$$

$$\beta = \frac{L^2+1}{2} \tag{10}$$

By Proposition 5.1, the regularized logistic regression loss function Equation (7) is $(\beta + \lambda)$-smooth. The above feature clipping technique is different from the DP-SGD algorithm [1] that only requires per-example gradient clipping. The major reason that we use data feature clipping (besides per-example gradient clipping), is for ensuring smoothness of the logistic regression loss function (by Proposition 5.1), which is a necessary condition for applying our privacy bound Theorem 3.3.

***Per-example Clipping on Unregularized Gradient (for reducing gradient sensitivity without harming smoothness or strong convexity).*** Although feature clipping already bounds the gradient sensitivity by $2\sqrt{2(L^2+1)}$ (by Proposition 5.1), this bound grows with the feature clipping norm $L$. This in turn restricts the signal to noise ratio, and tends to give suboptimal privacy-utility trade-off in practical experiments. Therefore, we additionally perform per-example $\ell_2$-clipping on the unregularized gradient (detailed pseudocode in Appendix F.1). Under per-example clipping on unregularized gradient, we prove in the following Proposition 5.2, that each gradient update in *regularized logistic regression* has finite gradient sensitivity, and preserves strong convexity and smoothness.

**Proposition 5.2.** *Let $\ell_0(\theta; \mathbf{x}, \mathbf{y})$ be the logistic regression loss function defined in Equation (8). Let $g_0(\theta; \mathbf{x}, \mathbf{y}) = \frac{\nabla \ell_0(\theta; \mathbf{x}, \mathbf{y})}{\|\nabla \ell_0(\theta; \mathbf{x}, \mathbf{y})\|_2} \cdot \min\{\|\nabla \ell_0(\theta; \mathbf{x}, \mathbf{y})\|_2, \frac{S_g}{2}\}$ be the clipped gradient of (unregularized) loss function $\ell_0(\theta; \mathbf{x}, \mathbf{y})$, under $\ell_2$ clipping norm $\frac{S_g}{2}$. If $g(\theta; \mathbf{x}, \mathbf{y}) = g_0(\theta; \mathbf{x}, \mathbf{y}) + \lambda\theta$, and if the data vector $\mathbf{x}$ has bounded $\ell_2$ norm, such that $\|\mathbf{x}\|_2 \leq L$, then $g(\theta; \mathbf{x}, y)$ has finite $\ell_2$-sensitivity $S_g$, is continuous, and is almost everywhere differentiable with*

$$\lambda \cdot \mathbb{I}_{(d+1)\cdot c} \preceq \nabla_\theta g(\theta; \mathbf{x}, \mathbf{y}) \preceq (\beta + \lambda) \cdot \mathbb{I}_{(d+1)\cdot c} \tag{11}$$

*for any $\theta, \theta' \in \mathbb{R}^{(d+1)\cdot c}$ and $\beta = \frac{L^2+1}{2}$.*

We provide complete proof for this proposition in Appendix F.4. This construction of clipped unregularized gradient enables us to enjoy the benefits of gradient clipping (such as for speeding up convergence [41, 14]) while satsifying the necessary smoothness and strong convexity conditions for applying our privacy dynamics bound.

## 5.2 Composition-based privacy bound and privacy dynamics analysis for DP-SGD

***Baseline composition-based privacy analysis.*** To the best of our knowledge, the moments accountant [1] combined with the privacy amplification bound for Subsampled Gaussian Mechanism [29] gives the strongest baseline composition-based privacy bound for DP-SGD. The original DP-SGD bound only holds for mini-batch sampling from a Poisson distribution. However, currently many privacy libraries [24, 40] still apply the DP-SGD analysis while implementing the "shuffle and partition" sampling scheme. Recent works [20, 21] further prove that the Rényi DP bound for DP-SGD under "shuffle and partition" matches that for DP-SGD under Poisson sampling in order. Therefore, we consider the moments accountant bound for subsampled Gaussian mechanisms [29] as a reasonable comparison baseline for our privacy bound of DP-SGD Algorithm 2 under "shuffle and partition".

***Improved privacy dynamics analysis.*** We now compute our privacy dynamics bound Theorem 4.2 for the modified DP-SGD algorithm Algorithm 2 under "shuffle and partition". By plugging the

notation transformation in Appendix F.1 into Theorem 4.2, we prove the following privacy dynamics theorem for noisy mini-batch gradient descent on regularized logistic regression.

**Corollary 5.3** (Privacy dynamics for noisy mini-batch gradient descent on regularized logistic regression). *For the regularized logistic regression loss Equation* (7) *with regularization coefficient* $\lambda$, *if the data feature vector is clipped in* $\ell_2$-*norm by* $L$, *and the unregularized gradient is clipped in* $\ell_2$ *norm by* $\frac{S_g}{2}$, *then for* $K \geq 1$ *and* $\frac{n}{b} \geq 2$, *Algorithm* 2 *with stepsize* $\eta < \frac{2}{(L^2+1)/2+2\lambda}$ *and noise multiplier* $\sigma_{mul}$ *satisfies* $(\alpha, \varepsilon_{norm} + \varepsilon)$-*Rényi Differential Privacy with*

$$\varepsilon \leq \varepsilon_0^{\lfloor \frac{n}{2b} \rfloor}(\alpha) \cdot \frac{1 - (1-\eta\lambda)^{2 \cdot (K-1) \cdot (n/b - \lfloor \frac{n}{2b} \rfloor)}}{1 - (1-\eta\lambda)^{2 \cdot (n/b - \lfloor \frac{n}{2b} \rfloor)}} + \frac{1}{\alpha - 1} \cdot \log \left( \underset{0 \leq j_0 < n/b}{Avg} \ e^{(\alpha-1)\varepsilon_0^{n/b - j_0}(\alpha)} \right) \quad (12)$$

*where the terms* $\varepsilon_0^j(\alpha)$ *is upper-bounded for any* $j = 1, \cdots, n/b$ *as follows.*

$$\varepsilon_0^j(\alpha) \leq \frac{2\alpha}{\sigma_{mul}^2} \cdot (1 - \eta\lambda)^{2 \cdot (j-1)} \cdot \frac{1}{\sum_{s=0}^{j-1}(1-\eta\lambda)^{2s}} \quad (13)$$

*Proof Sketch.* The proof is by applying Theorem 4.2 with gradient sensitivity $S_g$, smoothness constant $\frac{L^2+1}{2} + \lambda$, strong convexity constant $\lambda$ (derived by Proposition 5.2)) and noise standard deviation $\sigma = \sqrt{\frac{\eta}{2}} \cdot \frac{1}{b} \cdot \sigma_{mul} \cdot \frac{S_g}{2}$ (explained in Appendix F.1). $\square$

## 6 Conclusions and Discussion

We prove a novel converging last-iterate privacy bound for noisy stochastic mini-batch gradient descent on strongly convex smooth loss functions. Our bound substantially improves the prior privacy dynamics bound for noisy GD [15], by proving novel bounds for the additional privacy amplification (by randomized post-processing and sub-sampling) during training with stochastic mini-batches. Our results show that to obtain tighter privacy bound (thus achieving better privacy accuracy trade-off), differentially private learning algorithms needs to be evaluated by a last-iterate privacy bound, unless it has a very fast convergence (under which, due to the small number of epochs, the cost of composition bound is not significant).

***Future Work and Other Related Work***. For iteratively resampling a mini-batch of fixed size without replacement, there is room for improving our hidden-state privacy amplification bound Theorem 4.3 under smaller batch size $b$, because under *one* sampling without replacement step, the best known amplification rate is as small as $O(\frac{b^2}{n^2})$ [5, 37]. This suggests possibility for a better *hidden-state* privacy amplification bound (without using composition) for multiple steps of sub-sampling.

In our current analysis, strong convexity and smoothness of the loss functions are *necessary conditions* for obtaining a *converging* hidden-state Rényi DP bound. More specifically, strong convexity and smoothness ensure that log-Sobolev inequality with constant $c$ (which is a condition required by Lemma 3.2) holds throughout the training process (i.e. after any number of epochs $k$ and iterations $j$). A recent work Altschuler and Talwar [2] consider constrained optimization problem where $\theta$ is optimized over a bounded set with finite $\ell_2$ diameter (rather than the unconstrained parameter space $\mathbb{R}^d$ considered in this paper). Under such setting, they prove that convexity (instead of strong convexity) and smoothness of the loss functions suffice to enable a time-independent Rényi DP bound for noisy SGD. However, it remains an important open problem to further relax the convexity and smoothness conditions for proving converging Rényi DP bound.

One of the most important motivation for studying hidden-state Rényi DP bound in this paper, is that most State-of-the-art privacy accuracy trade-off for differentially private learning are achieved via the last-iterate model (instead of average of iterates). However, in terms of theoretical privacy utility trade-off, prior works [7, 8] prove that the average of iterates achieves optimal privacy-utility trade-off in certain regimes. For convex stochastic optimization, by exploiting the privacy amplification of last-iterate, recent works [19, 15, 22] prove that last-iterate also achieves asymptotically optimal privacy-utility trade-off (for appropriate choice of step sizes or continuous-time algorithm). Therefore, it remains an interesting open problem as to whether last-iterate or average of iterate enables better (non-assymptotic) privacy accuracy trade-off.

## Acknowledgments and Disclosure of Funding

The authors would like to thank Yaodong Yu, Chuan Guo, Maziar Sanjabi and anonymous reviewers for helpful discussions on drafts of this paper. This research is supported by Google PDPO faculty research award, Intel within the www.private-ai.org center, Meta faculty research award, the NUS Early Career Research Award (NUS ECRA award number NUS ECRA FY19 P16), and the National Research Foundation, Singapore under its Strategic Capability Research Centres Funding Initiative. Any opinions, findings and conclusions or recommendations expressed in this material are those of the author(s) and do not reflect the views of National Research Foundation, Singapore.

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
