# Appendix

## Table of Contents

# A  Symbols

**Algorithms and Definitions**

| Symbol | Meaning | Where |
|---|---|---|
| $\mathbf{x}$ | A data vector | Algorithm 1 |
| $n$ | Size of a dataset | Algorithm 1 |
| $b$ | Size of a mini-batch | Algorithm 1 |
| $K$ | Total number of epochs in a learning algorithm | Algorithm 1 |
| $\mathbb{I}_d$ | Identity matrix with $d$ rows and $d$ columns | Algorithm 1 |
| $\theta$ | Model parameters | Algorithm 1 |
| $\ell(\theta, \mathbf{x})$ | A loss function of $\theta$ parameterized by $\mathbf{x}$ | Algorithm 1 |
| $D, D'$ | Neighboring datasets that differ in at most one record | Appendix B |
| $S_g$ | $\ell_2$-sensitivity of total gradient $g(\theta; D) = \sum_{\mathbf{x} \in D} \nabla \ell(\theta; \mathbf{x})$ with regard to neighboring datasets. More specifically, $S_g = \max_{D, D', \theta} \|g(\theta, D) - g(\theta, D')\|_2$ | Algorithm 1 |
| $\mathcal{N}(\mu, \sigma \cdot \mathbb{I}_d)$ | Gaussian distribution over $\mathbb{R}^d$ with mean $\mu$ and covariance matrix $\sigma \cdot \mathbb{I}_d$ | Algorithm 1 |
| $\eta$ | Stepsize for each iterative update in the learning algorithm | Algorithm 1 |
| $\{0, 1, \ldots, n\}$ | The set of all integers between $0$ and $n$ | Algorithm 1 |
| $\mathbf{x}_i$ | Data point $i$ of dataset $D$, with indexing starting at 1 | Algorithm 1 |
| $\theta_k^j, B_k^j$ | The parameters or generated mini-batch after $k$ epochs and $j$ iterations of a learning algorithm, with indexing starting at 0 | Algorithm 1 |
| $\alpha$ | Rényi DP order | Appendix B |
| $(\varepsilon, \delta)$ | Differential Privacy parameters | Appendix B |
| $(\alpha, \varepsilon)$ | Rényi Differential Privacy parameters | Appendix B |
| $c$ | A log-Sobolev inequality constant | Appendix B |
| $\lambda$ | The strong convexity parameter for a loss function | Section 3 |
| $\beta$ | The smoothness parameter for a loss function | Section 3 |

**Probability and Information Theory**

| Symbol | Meaning | Where |
|---|---|---|
| $\mu$ | A distribution over $\mathbb{R}^d$ with density $\mu(\theta)$ | Section 3 |
| $p(\theta)$ | A probability density function over $\theta \in \mathbb{R}^d$ | Section 2 |
| $f_{\#}(\mu)$ | The push forward distribution of $\mu$ under mapping $f$ on the same domain | Section 3 |
| $\mu * \nu$ | The convolution of two distributions $\mu$ and $\nu$ | Section 3 |
| $R_\alpha(\mu\|\nu)$ | Rényi divergence between distributions $\mu$ and $\nu$ | Appendix B |
| $\mathbb{E}_{\theta\sim\mu}[f(\theta)]$ | Expectation of $f(\theta)$ with respect to $\mu(\theta)$ | Appendix B |
| $KL(\mu\|\nu)$ | Kullback-Leibler divergence between distributions $\mu$ and $\nu$ | Appendix D |

**Calculus and Linear Algebra**

| Symbol | Meaning |
|---|---|
| $t$ | A real scalar |
| $\lfloor t \rfloor$ | The largest integer that is smaller than or equal than a real number $t$ |
| $A_{i,j}$ | Element $i, j$ of matrix $A$ |
| $\dfrac{dy}{dx}$ | Derivative of $y$ with respect to $x$ |
| $\dfrac{\partial y}{\partial x}, \nabla_x y$ | Partial derivative of $y$ with respect to $x$ |
| $\nabla_x^2 y$ | Laplacian of $y$ with respect to $x$ |
| $\nabla_x y$ | Gradient of $y$ with respect to $x$ |
| $x \cdot y$ | Inner products between real vectors $x$ and $y$ |
| $x$ | $\ell_2$ norm of a vector $x$ |
| $A \otimes B$ | Tensor product between two real matrices $A$ and $B$ |
| $x^T$ | Transpose of a real vector or matrix $x$ |
| $f \circ g$ | Composition of the functions or mappings $f$ and $g$ |
| $A \preceq B$ | The matrix $B - A$ is semi-positive definite. |

# B  Preliminaries

**Definition B.1** (Differential Privacy [16, 17])**.** A randomized algorithm $\mathcal{A}$ is $(\varepsilon, \delta)$-differentially private if for any neighboring datasets $D, D'$, and for all possible event $S$ in the output space of $\mathcal{A}$,

$$P\left(\mathcal{A}(D) \in S\right) \leq e^\varepsilon \cdot P\left(\mathcal{A}(D') \in S\right) + \delta \tag{14}$$

where we say $D, D'$ are *neighboring* if they are of the *same size* and differ in at most one data record.

**Definition B.2** $((\alpha, \varepsilon)$-Rényi DP [28])**.** A randomized algorithm $\mathcal{A}$ is said to satisfy $(\alpha, \varepsilon)$-Rényi differential privacy (or $(\alpha, \varepsilon)$-Rényi DP for short), if for any neighboring datasets $D$ and $D'$,

$$R_\alpha(\mathcal{A}(D)\|\mathcal{A}(D')) \leq \varepsilon, \text{ where } R_\alpha(\mu\|\nu) = \frac{1}{\alpha - 1}\log\mathbb{E}_{\theta\sim\nu}\left[\left(\frac{\mu(\theta)}{\nu(\theta)}\right)^\alpha\right] \tag{15}$$

where $\mathcal{A}(D)$ $(\mathcal{A}(D'))$ denote the distribution of output given input dataset $D$ $(D')$, and $R_\alpha(\mu\|\nu)$ is the *Rényi divergence* [31] of order $\alpha > 1$ for two distributions with density $\mu(\theta)$ and $\nu(\theta)$ on $\mathbb{R}^d$.

**Definition B.3** (log-Sobolev Inequality [35]). A distribution $\nu$ over $\mathbb{R}^d$ satisfies the log-Sobolev inequality (LSI) with constant $c$ if for all smooth function $g : \mathbb{R}^d \to \mathbb{R}$ with $\mathbb{E}_{\theta \sim \nu}\left[g(\theta)^2\right] < \infty$,

$$\mathbb{E}_{\theta \sim \nu}\left[g(\theta)^2 \log\left(g(\theta)^2\right)\right] - \mathbb{E}_{\theta \sim \nu}\left[g(\theta)^2\right] \cdot \log \mathbb{E}_{\theta \sim \nu}\left[g(\theta)^2\right] \leq \frac{2}{c} \mathbb{E}_{\theta \sim \nu}\left[\|\nabla g(\theta)\|^2\right]. \quad (16)$$

## C  Discussion about the concurrent work [32, Corollary 3.3]

Chourasia et al. [15] prove privacy dynamics bound for noisy gradient descent, and Ryffel et al. [32] extend this bound to SGLD, by directly viewing each mini-batch update as gradient descent on a smaller dataset of size $b$ (which is the size of a mini-batch). This approach is similar to our approach for deriving the naive privacy dynamics baseline Theorem 2.1, and the expression in [32, Corollary 3.3] is very similar to the expression in our Theorem 2.1, except for having $n^2$ (instead of $b^2$) in the bound denominator.

However, an inspection of the proof for [32, Lemma 3.4] shows that, this difference between [32, Corollary 3.3] and our Theorem 2.1 is caused because by a flawed assumption in [32, Lemma 3.3]. More specifically, [32, Lemma 3.3] wrongly assume that the LSI constant proved in [15, Lemma 5] (which only holds for a GD process) would also similarly hold for a SGLD process that takes the form of a more complex mixture distribution. Here, each mixture component is the conditional distribution of last-iterate parameters given a fixed sequence of mini-batches.

This assumption is wrong because a given mixture distribution generally satisfies a different LSI constant than each of its component distributions. Moreover, bounding the LSI constant for a mixture distribution is largely an open problem [42, 43, 36, 6, 13]. The current best bound for this problem, to the best of our knowledge, is [13, Theorem 1], which says that the LSI constant for a mixture distribution, depends on the LSI constant for the distribution of each component, and the *worst-case* $\chi^2$ distance between *any* two components' distributions. Therefore, the actual LSI constant for SGLD process would be significantly smaller (related to the number of components in the mixture distribution) than the assumed LSI constant in [32, Lemma 3.4] (which only holds for the conditional parameter distribution given *a fixed mini-batch sequence*). After replacing the wrongly assumed LSI constant in [32, Lemma 3.4] with a correct LSI constant for SGLD process (that takes the form of mixture distribution), the privacy dynamics bounds in [32, Corollary 3.3] would be significantly worse (larger).

Due to this flawed assumption, in this paper, we do not compare our improved privacy dynamics theorem Theorem 3.3 with [32, Corollary 3.3]. Instead, we compare Theorem 3.3 with the naive privacy dynamics baseline Theorem 2.1 (which has similar expression as [32, Corollary 3.3]) in Figure 1.

## D  Proof for Section 3

We first establish a tool lemma for proving Lemma 3.1 and obtaining the partial differential inequality that bounds the growth of differential privacy loss.

**Lemma D.1** (Lemma 5 in Vempala and Wibisono [35]). *Suppose $\nu$ is a distribution that satisfies log-Sobolev inequality with constant $c > 0$, and that has smooth density $\mu(\theta)$. Let $\alpha \geq 1$. For all measure $\mu$ that has smooth density $\nu(\theta)$,*

$$\frac{I_\alpha(\mu\|\nu)}{E_\alpha(\mu\|\nu)} \geq \frac{2c}{\alpha^2} \cdot R_\alpha(\mu\|\nu) + \frac{2c}{\alpha^2} \cdot \alpha(\alpha-1)\frac{\partial R_\alpha(\mu\|\nu)}{\partial \alpha} \quad (17)$$

*where $I_\alpha(\mu\|\nu) = \mathbb{E}_{\theta \sim \nu}\left[\left(\frac{\mu(\theta)}{\nu(\theta)}\right)^\alpha \cdot \left\|\nabla \log \frac{\mu(\theta)}{\nu(\theta)}\right\|^2\right]$, and $E_\alpha(\mu\|\nu) = \mathbb{E}_{\theta \sim \nu}\left[\left(\frac{\mu(\theta)}{\nu(\theta)}\right)^\alpha\right]$*

*Proof.* This Lemma is initially proved in Vempala and Wibisono [35] Lemma 5. We give an alternative proof here, that only uses one step of inequality in Equation (23). We hope this alternative proof helps understand whether there is still room for improving this Lemma (e.g. by improving the inequality in Equation (23)).

We denote $\rho$ to be another distribution with density $\rho(\theta) = \frac{1}{E_\alpha(\mu\|\nu)} \cdot \left(\frac{\mu(\theta)}{\nu(\theta)}\right)^\alpha \cdot \nu(\theta)$. By simple integration, we verify that $\int \rho(\theta)d\theta = \frac{E_\alpha(\mu\|\nu)}{E_\alpha(\mu\|\nu)} = 1$. By definition,

$$\frac{I_\alpha(\mu\|\nu)}{E_\alpha(\mu\|\nu)} = \frac{\mathbb{E}_{\theta\sim\nu}\left[\left(\frac{\mu(\theta)}{\nu(\theta)}\right)^\alpha \cdot \left\|\nabla \log \frac{\mu(\theta)}{\nu(\theta)}\right\|^2\right]}{E_\alpha(\mu\|\nu)} \tag{18}$$

$$= \mathbb{E}_{\theta\sim\nu}\left[\frac{\rho(\theta)}{\nu(\theta)} \cdot \left\|\nabla \log \frac{\mu(\theta)}{\nu(\theta)}\right\|^2\right] \tag{19}$$

$$= \frac{1}{\alpha^2}\mathbb{E}_{\theta\sim\nu}\left[\frac{\rho(\theta)}{\nu(\theta)} \cdot \left\|\nabla \log \left(\frac{\mu(\theta)}{\nu(\theta)}\right)^\alpha\right\|^2\right] \tag{20}$$

by $\nabla \log E_\alpha(\mu\|\nu) = 0$, $\quad = \frac{1}{\alpha^2}\mathbb{E}_{\theta\sim\nu}\left[\frac{\rho(\theta)}{\nu(\theta)} \cdot \left\|\nabla \log \left(\frac{\mu(\theta)}{\nu(\theta)}\right)^\alpha - \nabla \log E_\alpha(\mu\|\nu)\right\|^2\right] \tag{21}$

$$= \frac{1}{\alpha^2}\mathbb{E}_{\theta\sim\nu}\left[\frac{\rho(\theta)}{\nu(\theta)} \cdot \left\|\nabla \log \frac{\rho(\theta)}{\nu(\theta)}\right\|^2\right] \tag{22}$$

By definition, $\mathbb{E}_{\theta\sim\nu}\left[\frac{\rho(\theta)}{\nu(\theta)} \cdot \left\|\nabla \log \frac{\rho(\theta)}{\nu(\theta)}\right\|^2\right]$ is the relative Fisher information $J(\rho\|\nu)$ of $\rho$ with respect to $\nu$. A celebrated equivalence result (e.g. Section 2.2 of Vempala and Wibisono [35]) says that, if and only if $\nu$ satsifies log-Sobolev inequality with constant $c$, the following relation between the KL divergence and relative Fisher information holds for all $\rho$:

$$KL(\rho\|\nu) \le \frac{1}{2c}J(\rho\|\nu) \tag{23}$$

By plugging Equation (23) into Equation (22), we prove

$$\frac{I_\alpha(\mu\|\nu)}{E_\alpha(\mu\|\nu)} \ge \frac{2c}{\alpha^2} \cdot KL(\rho\|\nu) \tag{24}$$

by definition, $\quad = \frac{2c}{\alpha^2} \cdot \int_{\mathbb{R}^n} \rho(\theta) \log \frac{\rho(\theta)}{\nu(\theta)} d\theta \tag{25}$

by definition of $\rho$, $\quad = \frac{2c}{\alpha^2} \cdot \mathbb{E}_{\theta\sim\nu}\left[\frac{1}{E_\alpha(\mu\|\nu)} \cdot \left(\frac{\mu(\theta)}{\nu(\theta)}\right)^\alpha \cdot \left(\alpha \log \frac{\mu(\theta)}{\nu(\theta)} - \log E_\alpha(\mu\|\nu)\right)\right] \tag{26}$

$$= \frac{2c}{\alpha} \cdot \frac{1}{E_\alpha(\mu\|\nu)} \cdot \mathbb{E}_{\theta\sim\nu}\left[\frac{\partial}{\partial\alpha}\left(\frac{\mu(\theta)}{\nu(\theta)}\right)^\alpha\right] - \frac{2c}{\alpha^2} \cdot \frac{\log E_\alpha(\mu\|\nu)}{E_\alpha(\mu\|\nu)} \cdot \mathbb{E}_{\theta\sim\nu}\left[\left(\frac{\mu(\theta)}{\nu(\theta)}\right)^\alpha\right] \tag{27}$$

$$= \frac{2c}{\alpha} \cdot \frac{1}{E_\alpha(\mu\|\nu)} \cdot \mathbb{E}_{\theta\sim\nu}\left[\frac{\partial}{\partial\alpha}\left(\frac{\mu(\theta)}{\nu(\theta)}\right)^\alpha\right] - \frac{2c}{\alpha^2} \log E_\alpha(\mu\|\nu) \tag{28}$$

By exchanging the order of derivative and expectation (because $\mu(\theta)$ and $\nu(\theta)$ are smooth densities), we prove that Equation (28) is equivalent to the following inequality.

$$\frac{I_\alpha(\mu\|\nu)}{E_\alpha(\mu\|\nu)} \ge \frac{2c}{\alpha} \cdot \frac{\partial}{\partial\alpha} \log \mathbb{E}_\alpha(\mu\|\nu) - \frac{2c}{\alpha^2} \log E_\alpha(\mu\|\nu) \tag{29}$$

By definition, we have $E_\alpha(\mu\|\nu) = e^{(\alpha-1)R_\alpha(\mu\|\nu)}$, therefore we prove

$$\frac{I_\alpha(\mu\|\nu)}{E_\alpha(\mu\|\nu)} \ge \frac{2c}{\alpha} \cdot \left(R_\alpha(\mu\|\nu) + (\alpha-1)\frac{\partial R_\alpha(\mu\|\nu)}{\partial\alpha}\right) - \frac{2c}{\alpha^2} \cdot (\alpha-1)R_\alpha(\mu\|\nu) \tag{30}$$

$$= \frac{2c}{\alpha^2} \cdot R_\alpha(\mu\|\nu) + \frac{2c}{\alpha^2} \cdot \alpha(\alpha-1)\frac{\partial R_\alpha(\mu\|\nu)}{\partial\alpha} \tag{31}$$

$\square$

### D.1 Proof for Lemma 3.1

*Lemma 3.1.* Let $\mu, \nu$ be two distributions on $\mathbb{R}^d$. Let $f : \mathbb{R}^d \to \mathbb{R}^d$ be a measurable mapping on $\mathbb{R}^d$. We denote $\mathcal{N}(0, 2t\sigma^2 \cdot \mathbb{I}_d)$ to be the standard Gaussian distribution on $\mathbb{R}^d$ with covariance

matrix $2t\sigma^2 \cdot \mathbb{I}_d$. We denote $p_t(\theta)$ and $p'_t(\theta)$ to be the probability density functions for the distributions $f_\#(\mu) * \mathcal{N}(0, 2t\sigma^2\mathbb{I}_d)$ and $f_\#(\nu) * \mathcal{N}(0, 2t\sigma^2\mathbb{I}_d)$ respectively, where $f_\#(\mu), f_\#(\nu)$ denote the push forward distributions of $\mu, \nu$ under mapping $f$. Then if $\mu$ and $\nu$ satisfy log-Sobolev inequality with constant $c$, and if the mapping $f$ is $L$-Lipschitz, then for any order $\alpha > 1$,

$$\frac{\partial}{\partial t} R_\alpha\left(p_t(\theta) \| p'_t(\theta)\right) \leq -c_t \cdot 2\sigma^2 \cdot \left(\frac{R_\alpha(p_t(\theta)\|p'_t(\theta))}{\alpha} + (\alpha - 1) \cdot \frac{\partial}{\partial \alpha} R_\alpha(p_t(\theta)\|p'_t(\theta))\right), \quad (32)$$

where $c_t = \left(\frac{L^2}{c} + 2t\sigma^2\right)^{-1}$ is the log-Sobolev inequality constant for distributions $p_t(\theta)$ and $p'_t(\theta)$.

*Proof.* By definition, $p_t(\theta)$ and $p'_t(\theta)$ are probability density functions for distributions $f_\#(\mu) *\mathcal{N}(0, 2t\sigma^2\mathbb{I}_d)$ and $f_\#(\nu)*\mathcal{N}(0, 2t\sigma^2\mathbb{I}_d)$ respectively. Therefore $p_t(\theta)$ and $p'_t(\theta)$ satisfy the following Fokker-Planck equations.

$$\frac{\partial p_t(\theta)}{\partial t} = \sigma^2 \Delta p_t(\theta), \quad (33)$$

$$\frac{\partial p'_t(\theta)}{\partial t} = \sigma^2 \Delta p'_t(\theta) \quad (34)$$

We denote $E_\alpha(p_t(\theta)\|p'_t(\theta)) = \int \frac{p_t(\theta)^\alpha}{p'_t(\theta)^{\alpha-1}} d\theta$ to be the moment of the likelihood ratio function. Then by definition of Rényi divergence, we prove

$$R_\alpha(p_t(\theta)\|p'_t(\theta)) = \frac{1}{\alpha - 1} \log E_\alpha(p_t(\theta)\|p'_t(\theta)) \quad (35)$$

Therefore we compute the rate of Rényi divergence with regard to $t$ as follows.

$$\frac{\partial}{\partial t} R_\alpha(p_t(\theta)\|p'_t(\theta)) = \frac{1}{\alpha - 1} \frac{\partial}{\partial t} \log E_\alpha(p_t(\theta)\|p'_t(\theta)) \quad (36)$$

$$= \frac{1}{(\alpha - 1)E_\alpha(p_t(\theta)\|p'_t(\theta))} \cdot \frac{\partial}{\partial t} E_\alpha(p_t(\theta)\|p'_t(\theta)) \quad (37)$$

By definition of $E_\alpha(p_t(\theta)\|p'_t(\theta))$, $\quad = \frac{1}{(\alpha - 1)E_\alpha(p_t(\theta)\|p'_t(\theta))} \cdot \frac{\partial}{\partial t}\left(\int \frac{p_t(\theta)^\alpha}{p'_t(\theta)^{\alpha-1}} d\theta\right) \quad (38)$

By exchanging the order of derivative and integration, we prove

$$\frac{\partial}{\partial t} R_\alpha(p_t(\theta)\|p'_t(\theta)) = \frac{1}{(\alpha - 1)E_\alpha(p_t(\theta)\|p'_t(\theta))} \cdot \int \frac{\partial}{\partial t} \frac{p_t(\theta)^\alpha}{p'_t(\theta)^{\alpha-1}} d\theta \quad (39)$$

$$= \frac{1}{(\alpha - 1)E_\alpha(p_t(\theta)\|p'_t(\theta))} \cdot \int \left(\alpha \cdot \frac{p_t(\theta)^{\alpha-1}}{p'_t(\theta)^{\alpha-1}} \cdot \frac{\partial p_t(\theta)}{\partial t} - (\alpha - 1) \cdot \frac{p_t(\theta)^\alpha}{p'_t(\theta)^\alpha} \cdot \frac{\partial p'_t(\theta)}{\partial t}\right) d\theta \quad (40)$$

By the Fokker-Planck equations Equation (33) and Equation (34), we substitute the terms $\frac{\partial p_t(\theta)}{\partial t}$ and $\frac{\partial p'_t(\theta)}{\partial t}$ in the above equation as follows.

$$\frac{\partial}{\partial t} R_\alpha(p_t(\theta)\|p'_t(\theta)) \quad (41)$$

$$= \frac{1}{(\alpha - 1)E_\alpha(p_t(\theta)\|p'_t(\theta))} \cdot \int \left(\alpha\sigma^2 \cdot \frac{p_t(\theta)^{\alpha-1}}{p'_t(\theta)^{\alpha-1}} \cdot \Delta p_t(\theta) - (\alpha - 1)\sigma^2 \cdot \frac{p_t(\theta)^\alpha}{p'_t(\theta)^\alpha} \cdot \Delta p'_t(\theta)\right) d\theta \quad (42)$$

By applying Green's first identity in Equation (42), the first intergration term in Equation (42) is changed to

$$\int \alpha\sigma^2 \cdot \frac{p_t(\theta)^{\alpha-1}}{p'_t(\theta)^{\alpha-1}} \cdot \Delta p_t(\theta) d\theta = \lim_{r\to\infty} \int_{B_r} \alpha\sigma^2 \cdot \frac{p_t(\theta)^{\alpha-1}}{p'_t(\theta)^{\alpha-1}} \cdot \Delta p_t(\theta) d\theta \quad (43)$$

$$= \lim_{r\to\infty} \int_{\partial B_r} \alpha\sigma^2 \cdot \frac{p_t(\theta)^{\alpha-1}}{p'_t(\theta)^{\alpha-1}} \cdot \nabla p_t(\theta) \cdot d\mathbf{S} - \int \nabla\left(\alpha\sigma^2 \cdot \frac{p_t(\theta)^{\alpha-1}}{p'_t(\theta)^{\alpha-1}}\right) \cdot \nabla p_t(\theta) d\theta, \quad (44)$$

where $B_r$ is the unit ball centered around origin in $d$-dimensional Euclidean space with radius $r$. The limits in the first term of Equation (44) becomes zero given the smoothness and fast decay properties of $p_t(\theta)$, and the Lebesgue integrability of $\frac{p_t(\theta)^{\alpha-1}}{p_t'(\theta)^{\alpha-1}} \cdot p_t(\theta)$. Therefore we prove that

$$\int \alpha\sigma^2 \cdot \frac{p_t(\theta)^{\alpha-1}}{p_t'(\theta)^{\alpha-1}} \cdot \Delta p_t(\theta)d\theta = -\int \nabla\left(\alpha\sigma^2 \cdot \frac{p_t(\theta)^{\alpha-1}}{p_t'(\theta)^{\alpha-1}}\right) \cdot \nabla p_t(\theta)d\theta, \qquad (45)$$

Similarly by applying Green's first identity in Equation (42), the second intergration term in Equation (42) is changed to

$$\int -(\alpha-1)\sigma^2 \cdot \frac{p_t(\theta)^{\alpha}}{p_t'(\theta)^{\alpha}} \cdot \Delta p_t'(\theta)d\theta = \int \nabla\left((\alpha-1)\sigma^2 \cdot \frac{p_t(\theta)^{\alpha}}{p_t'(\theta)^{\alpha}}\right) \cdot \nabla p_t'(\theta)d\theta, \qquad (46)$$

(This techinque for using Green's identity and bounding the rate of entropy change with relative Fisher information-like quantity, has been previously used for KL divergence in **?** ], and Rényi divergence in Vempala and Wibisono [35], Chourasia et al. [15]. The result is also closely related to the well known *de Bruijn's indentity* in information theory literature. )

By plugging in Equation (45) and Equation (46) into Equation (42), we prove that

$$\frac{\partial}{\partial t}R_\alpha(p_t(\theta)\|p_t'(\theta)) \qquad (47)$$

$$= \frac{\sigma^2}{(\alpha-1)E_\alpha(p_t(\theta)\|p_t'(\theta))} \cdot \int -\alpha \cdot \left\langle \nabla\left(\frac{p_t(\theta)^{\alpha-1}}{p_t'(\theta)^{\alpha-1}}\right), \nabla p_t(\theta)\right\rangle + (\alpha-1) \cdot \left\langle \nabla\left(\frac{p_t(\theta)^{\alpha}}{p_t'(\theta)^{\alpha}}\right), \nabla p_t'(\theta)\right\rangle d\theta \qquad (48)$$

$$= \frac{\alpha(\alpha-1)\sigma^2}{(\alpha-1)E_\alpha(p_t(\theta)\|p_t'(\theta))} \cdot \int \frac{p_t(\theta)^{\alpha-2}}{p_t'(\theta)^{\alpha-2}} \cdot \left\langle \nabla\left(\frac{p_t(\theta)}{p_t'(\theta)}\right), -\nabla p_t(\theta) + \frac{p_t(\theta)}{p_t'(\theta)}\nabla p_t'(\theta)\right\rangle d\theta \quad (49)$$

$$= -\frac{\alpha\sigma^2}{E_\alpha(p_t(\theta)\|p_t'(\theta))} \cdot \int \frac{p_t(\theta)^{\alpha-2}}{p_t'(\theta)^{\alpha-2}} \cdot \left\langle \nabla\left(\frac{p_t(\theta)}{p_t'(\theta)}\right), \nabla\left(\frac{p_t(\theta)}{p_t'(\theta)}\right)\right\rangle \cdot p_t'(\theta)d\theta \qquad (50)$$

$$= -\alpha\sigma^2 \cdot \frac{I_\alpha(p_t(\theta)\|p_t'(\theta))}{E_\alpha(p_t(\theta)\|p_t'(\theta))}, \qquad (51)$$

where $I_\alpha(p_t(\theta)\|p_t'(\theta)) = \int \frac{p_t(\theta)^{\alpha-2}}{p_t'(\theta)^{\alpha-2}} \cdot \left\langle \nabla\left(\frac{p_t(\theta)}{p_t'(\theta)}\right), \nabla\left(\frac{p_t(\theta)}{p_t'(\theta)}\right)\right\rangle \cdot p_t'(\theta)d\theta$. Therefore, if $p_t(\theta)$ and $p_t'(\theta)$ satisfy log-Sobolev inequality with constant $c_t$, then by Lemma D.1, we obtain the following inequality.

$$\frac{I_\alpha(p_t(\theta)\|p_t'(\theta))}{E_\alpha(p_t(\theta)\|p_t'(\theta))} \geq \frac{2c_t}{\alpha^2}R_\alpha(p_t(\theta)\|p_t'(\theta)) + \frac{2c_t}{\alpha^2} \cdot \alpha(\alpha-1)\frac{\partial}{\partial\alpha}R_\alpha(p_t(\theta)\|p_t'(\theta)) \qquad (52)$$

Meanwhile, by Lemma 16 in Vempala and Wibisono [35] and Lemma 17 in Vempala and Wibisono [35], the distributions (with densities $p_t(\theta)$ and $p_t'(\theta)$) indeed satisfy $c_t$-log-Sobolev inequality with

$$c_t = \left(\frac{L^2}{c} + 2t\sigma^2\right)^{-1} \qquad (53)$$

By plugging Equation (53) and Equation (52) into Equation (51), we prove the following bound for the rate of Rényi divergence.

$$\frac{\partial}{\partial t}R_\alpha(p_t(\theta)\|p_t'(\theta)) \leq -\alpha\sigma^2 \cdot \left(\frac{2c_t}{\alpha^2}R_\alpha(p_t(\theta)\|p_t'(\theta)) + \frac{2c_t}{\alpha^2} \cdot \alpha(\alpha-1) \cdot \frac{\partial}{\partial\alpha}R_\alpha(p_t(\theta)\|p_t'(\theta))\right) \qquad (54)$$

$$= -c_t \cdot 2\sigma^2 \cdot \left(\frac{R_\alpha(p_t(\theta)\|p_t'(\theta))}{\alpha} + (\alpha-1) \cdot \frac{\partial}{\partial\alpha}R_\alpha(p_t(\theta)\|p_t'(\theta))\right) \qquad (55)$$

where $c_t = \left(\frac{L^2}{c} + 2t\sigma^2\right)^{-1}$ is the log-Sobolev inequality constant for distributions $p_t(\theta)$ and $p_t'(\theta)$. $\qquad\square$

## D.2 Proof for Lemma 3.2

***Reduce analysis to point initialization:***. Without loss of generality, in this paper, we only analyze recursive Rényi DP bounds for Algorithm 1 under an arbitrary point initialization for initial parameters $\theta_0^0$. This is because, under an arbitrary initialization distribution, the last-iterate parameters $\theta_K^0$ in Algorithm 1 follow a mixture distribution, with each component being the conditional output distribution given fixed initial parameters $\theta_0^0$. Therefore, by the quasi-convexity of Rényi divergence, the largest (worst-case) Rényi DP bound for $R_\alpha(p(\theta_K^0|\theta_0^0)\|p(\theta'^0_K|\theta_0^0))$ over all possible initial parameters $\theta_0^0$, is also an upper bound for the Rényi privacy loss $R_\alpha(\theta_K^0\|\theta'^0_K)$ between (mixture) last-iterate parameters distributions for running Algorithm 1 on two neighboring datasets.

We now proceed to prove the recursive privacy bound Lemma 3.2.

*Lemma 3.2.* Let $D, D'$ be an arbitrary pair of neighboring datasets that differ in the $i_0$-th data point (i.e. $x_{i_0} \neq x'_{i_0}$). Let $B_k^j$ be a fixed mini-batch used (in iteration $j$ of epoch $k$) in Algorithm 1, which contains $b$ indices sampled from $\{1, \cdots, n\}$. We denote $\theta_k^j$ and $\theta'^j_k$ as the intermediate parameters in Algorithm 1 on input datasets $D$ and $D'$, respectively. If the distributions of $\theta_k^j$ and $\theta'^j_k$ satisfy log-Sobolev inequality with a constant $c$, and if the mini-batch GD mapping $f(\theta) = \theta - \eta \cdot \frac{1}{b} \cdot \sum_{i \in B_k^j} \ell(\theta; \mathbf{x}_i)$ is $L$-Lipschitz for parameters $\theta$, then the following recursive bound for Rényi divergence holds.

$$\frac{R_\alpha(\theta_k^{j+1}\|\theta'^{j+1}_k)}{\alpha} \leq \begin{cases} \frac{R_{\alpha'}(\theta_k^j\|\theta'^j_k)}{\alpha'} \cdot \left(1 + \frac{c \cdot 2\eta\sigma^2}{L^2}\right)^{-1} & \text{if } i_0 \notin B_k^j \\ \frac{R_\alpha(\theta_k^j\|\theta'^j_k)}{\alpha} + \frac{\eta S_g^2}{4\sigma^2 b^2} & \text{if } i_0 \in B_k^j \end{cases} \text{ with } \alpha' = \frac{\alpha - 1}{1 + \frac{c \cdot 2\eta\sigma^2}{L^2}} + 1.$$

(56)

*Proof.*       1. When $i_0 \notin B_k^j$, the noisy mini-batch gradient descent mapping under both dataset $D$ and $D'$ is written in the same way as $f(\theta) + \mathcal{N}(0, 2\eta\sigma^2 \mathbb{I}_d)$, where $f(\theta) = \theta - \frac{\eta}{b} \cdot \sum_{i \in B_k^j} \nabla\ell(\theta; \mathbf{x}_i)$ (this is because $x_i = x'_i$ for $i \in B_k^j$, when $i_0 \notin B_k^j$). Therefore, we could use Lemma 3.1 and solve Equation (1) on the interval $t \in [0, \eta]$ to obtain the recursive privacy bound, where the Rényi divergence at $t = 0$ (i.e. before the update in iteration $j$ of epoch $k$ takes place) satisfies $R_\alpha(p_0(\theta)\|p'_0(\theta)) \leq \varepsilon_k^j(\alpha)$. We denote function $R(\alpha, t) = R_\alpha(p_t(\theta)\|p'_t(\theta))$. Then Equation (1) is equivalent to the following equation.

$$\begin{cases} \frac{\partial}{\partial t}R(\alpha, t) \leq -\left(\frac{L^2}{c} + 2t\sigma^2\right)^{-1} \cdot 2\sigma^2 \cdot \left(\frac{R(\alpha,t)}{\alpha} + (\alpha - 1) \cdot \frac{\partial}{\partial \alpha}R(\alpha, t)\right), \\ R(\alpha, 0) \leq \varepsilon_k^j(\alpha) \end{cases}$$

(57)

By substituting $u(\alpha, t) = \frac{R(\alpha,t)}{\alpha}$ into Equation (57), we prove that Equation (57) is equivalent to the following equation.

$$\begin{cases} \frac{\partial}{\partial t}u(\alpha, t) \leq -\left(\frac{L^2}{c} + 2t\sigma^2\right)^{-1} \cdot 2\sigma^2 \cdot \left(u(\alpha, t) + (\alpha - 1) \cdot \frac{\partial}{\partial \alpha}u(\alpha, t)\right), \\ u(\alpha, 0) \leq \frac{\varepsilon_k^j(\alpha)}{\alpha} \end{cases}$$

(58)

By change of variable $y = \ln(\alpha - 1)$, we prove that Equation (58) is equivalent to the following equation.

$$\begin{cases} \frac{\partial}{\partial t}u(y, t) \leq -\left(\frac{L^2}{c} + 2t\sigma^2\right)^{-1} \cdot 2\sigma^2 \cdot \left(u(y, t) + \frac{\partial}{\partial y}u(y, t)\right), \\ u(y, 0) \leq \frac{\varepsilon_k^j(e^y + 1)}{e^y + 1} \end{cases}$$

(59)

Now we do change of variable $\begin{cases} \tau = t \\ z = y - \int_0^t \left(\frac{L^2}{c} + 2t'\sigma^2\right)^{-1} \cdot 2\sigma^2 dt' + \ln\left(1 + \frac{c \cdot 2\eta\sigma^2}{L^2}\right) \end{cases}$

then by chain rule, we prove the following expressions for the partial derivatives using new

variables.

$$
\begin{cases}
\frac{\partial u}{\partial t} = \frac{\partial u}{\partial \tau} \cdot \frac{\partial \tau}{\partial t} + \frac{\partial u}{\partial z} \cdot \frac{\partial z}{\partial t} = \frac{\partial u}{\partial \tau} - \frac{\partial u}{\partial z} \cdot \left(\frac{L^2}{c} + 2t\sigma^2\right)^{-1} \cdot 2\sigma^2 \\
\frac{\partial u}{\partial y} = \frac{\partial u}{\partial \tau} \cdot \frac{\partial \tau}{\partial y} + \frac{\partial u}{\partial z} \cdot \frac{\partial z}{\partial y} = \frac{\partial u}{\partial z}
\end{cases}
\tag{60}
$$

By plugging Equation (60) into Equation (59), we prove that the partial differential inequality 59 is equivalent to the following inequality under new variables $\tau$ and $z$.

$$
\begin{cases}
\frac{\partial}{\partial \tau} u(z,\tau) \leq -\left(\frac{L^2}{c} + 2\tau\sigma^2\right)^{-1} \cdot 2\sigma^2 \cdot u(z,\tau), \\
u(z,0) \leq \dfrac{\varepsilon_k^j \left(e^{z - \ln\left(1 + \frac{c \cdot 2\eta\sigma^2}{L^2}\right)} + 1\right)}{e^{z - \ln\left(1 + \frac{c \cdot 2\eta\sigma^2}{L^2}\right)} + 1}
\end{cases}
\tag{61}
$$

Now we observe that given any fixed $z$, Equation (61) is an ordinary differential equation with regard to $\tau$, with a decay term that proportional to $-\left(\frac{L^2}{c} + 2\tau\sigma^2\right)^{-1} \cdot 2\sigma^2 \cdot u(z,\tau)$. Therefore, we directly solve Equation (61), and prove that

$$
\ln(u(z,\tau)) - \ln(u(z,0)) \leq -\int_0^\tau \left(\frac{L^2}{c} + 2\tau'\sigma^2\right)^{-1} \cdot 2\sigma^2 d\tau' = -\ln\left(1 + \frac{c \cdot 2\tau\sigma^2}{L^2}\right)
\tag{62}
$$

We take $\tau = \eta$ in Equation (62), then we prove

$$
\ln u(z,\eta) - \ln u(z,0) \leq -\ln\left(1 + \frac{c \cdot 2\eta\sigma^2}{L^2}\right)
\tag{63}
$$

By plugging the initial condition for $u(z,0)$ in Equation (61) into Equation (63), we prove that the solution for $u$ at the end of a set $\tau = \eta$ satisfies the following inequality.

$$
u(z,\eta) \leq \dfrac{\varepsilon_k^j \left(e^{z - \ln\left(1 + \frac{c \cdot 2\eta\sigma^2}{L^2}\right)} + 1\right)}{e^{z - \ln\left(1 + \frac{c \cdot 2\eta\sigma^2}{L^2}\right)} + 1} \cdot \left(1 + \frac{c \cdot 2\eta\sigma^2}{L^2}\right)^{-1}
\tag{64}
$$

Now we translate the variables $z = z, \tau = \eta$ back to the old variables $y$ and $t$ by definitions, and prove that $t = \tau = \eta$ and $z = y - \int_0^t \left(\frac{L^2}{c} + 2t'\sigma^2\right)^{-1} \cdot 2\sigma^2 dt' + \ln\left(1 + \frac{c \cdot 2\eta\sigma^2}{L^2}\right) = y$. Therefore, under these variable substitutions, we prove that Equation (63) is equivalent to the following equation.

$$
u(y,\eta) \leq \dfrac{\varepsilon_k^j \left(e^{y - \ln\left(1 + \frac{c \cdot 2\eta\sigma^2}{L^2}\right)} + 1\right)}{e^{y - \ln\left(1 + \frac{c \cdot 2\eta\sigma^2}{L^2}\right)} + 1} \cdot \left(1 + \frac{c \cdot 2\eta\sigma^2}{L^2}\right)^{-1}
\tag{65}
$$

Finally, we translate the variable $y$ back to $\alpha$, under the fixed variable $t = \eta$, by the definition $y = \log(\alpha - 1)$. Therefore, we prove that Equation (65) is equivalent to the following solution for $u(\alpha, \eta)$.

$$
u(\alpha, \eta) \leq \dfrac{\varepsilon_k^j \left((\alpha - 1) \cdot \left(1 + \frac{c \cdot 2\eta\sigma^2}{L^2}\right)^{-1} + 1\right)}{(\alpha - 1) \cdot \left(1 + \frac{c \cdot 2\eta\sigma^2}{L^2}\right)^{-1} + 1} \cdot \left(1 + \frac{c \cdot 2\eta\sigma^2}{L^2}\right)^{-1}
\tag{66}
$$

By the definition $u(\alpha, t) = \frac{R(\alpha, t)}{\alpha}$, we prove that Equation (66) is equivalent to the following solution for $R(\alpha, t)$.

$$
\dfrac{R(\alpha, \eta)}{\alpha} \leq \dfrac{\varepsilon_k^j \left((\alpha - 1) \cdot \left(1 + \frac{c \cdot 2\eta\sigma^2}{L^2}\right)^{-1} + 1\right)}{(\alpha - 1) \cdot \left(1 + \frac{c \cdot 2\eta\sigma^2}{L^2}\right)^{-1} + 1} \cdot \left(1 + \frac{c \cdot 2\eta\sigma^2}{L^2}\right)^{-1}
\tag{67}
$$

Therefore, by using the definition $R(\alpha, \eta) = \varepsilon_k^{j+1}(\alpha)$, we finish the proof for the Lemma statement, that when $i_0 \neq B_k^j$,

$$\frac{R_\alpha(\theta_k^{j+1}\|\theta'^{j+1}_k)}{\alpha} \leq \frac{R_{\alpha'}(\theta_k^j\|\theta'^j_k)}{\alpha'} \cdot \left(1 + \frac{c \cdot 2\eta\sigma^2}{L^2}\right)^{-1} \tag{68}$$

where $\alpha' = (\alpha - 1) \cdot \left(1 + \frac{c \cdot 2\eta\sigma^2}{L^2}\right)^{-1} + 1$.

2. When $i_0 \in B_k^j$, by composition theorem for Rényi differential privacy [28], and by the Rényi privacy bound for Gaussian mechanism [28] under $\ell_2$-sensitivity $S_g/b$ (for batch averaged gradient) and noise $\mathcal{N}(0, 2\eta\sigma^2\mathbb{I}_d)$, we prove that

$$R_\alpha(\theta_k^{j+1}\|\theta'^{j+1}_k) \leq R_\alpha(\theta_k^j, \theta_k^{j+1}\|\theta'^j_k, \theta'^{j+1}_k) \tag{69}$$

$$\leq R_\alpha(\theta_k^j\|\theta'^j_k) + \frac{\eta^2(S_g/b)^2}{2 \cdot 2\eta\sigma^2} = R_\alpha(\theta_k^j\|\theta'^j_k) + \frac{\alpha\eta S_g^2}{4\sigma^2 b^2}. \tag{70}$$

$\square$

## D.3  Proof for LSI sequence for noisy mini-batch gradient descent

To apply the recursive privacy bound Lemma 3.2 and prove a converging privacy dynamics, we first need to prove that the distributions of parameters $\theta_k^j$ in Algorithm 1 satisfy log-Sobolev inequality with certain constant $c_k^j$, that depends on $k$ and $j$. In this section, we prove that noisy mini-batch gradient descent on convex smooth loss function, as well as strongly convex smooth loss functions, satisfies LSI with certain sequences of constants as follows.

**Lemma D.2** (LSI constant sequence in Algorithm 1 for convex smooth loss). *Suppose that the loss function $\ell(\theta; \mathbf{x})$ in Algorithm 1 is convex and $\beta$-smooth. If the step-size $\eta < \frac{2}{\beta}$, then for any $k = 0, \cdots, K-1$ and $j = 0, \cdots, n/b - 1$, the distribution of parameters $\theta_k^j$ in Algorithm 1 satisfies $c_k^j$-log-Sobolev inequality with*

$$c_k^j = \frac{1}{2\eta\sigma^2 \cdot (k \cdot n/b + j)}, \tag{71}$$

*and we define $c_0^0 = \frac{1}{0} = +\infty$.*

*Proof.* The mini-batch noisy gradient descent update could be written as $\theta_k^{j+1} = f(\theta_k^j) + \mathcal{N}(0, 2\eta\sigma^2\mathbb{I}_d)$, where $f$ is a deterministic mapping on $\mathbb{R}^d$ written as $f(\theta) = \theta - \eta \cdot \frac{\sum_{i \in B} \ell(\theta; \mathbf{x}_i)}{b}$, and $B$ is a mini-batch of size $b$ consisting of indices selected from $0, \cdots, n$.

Because the initialization is point distribution around $\theta_0$, therefore $\theta_0^0$ satisfies log-Sobolev inequality with constant $c_0 = \infty$.

By the $\beta$-smoothness of $\ell(\theta; \mathbf{x})$, and by $\eta < \frac{2}{\beta}$, we prove that the mini-batch gradient mapping $f(\theta)$ is 1-Lipschitz. Further using LSI under Lipchitz mapping (Lemma 16 in Vempala and Wibisono [35]) and under Gaussian convolution (Lemma 17 in Vempala and Wibisono [35]), we prove that

$$\frac{1}{c_k^j} = \frac{1}{c_k^{j-1}} + 2\eta\sigma^2 \tag{72}$$

$$= \frac{1}{c_0^0} + 2\eta\sigma^2 \cdot (k \cdot n/b + j) \tag{73}$$

$$\text{by } c_0^0 = +\infty, \quad = 2\eta\sigma^2 \cdot (k \cdot n/b + j) \tag{74}$$

$$\tag{75}$$

This suffices to prove the LSI sequence in Equation (71). $\square$

Similarly, we prove another LSI sequence for noisy mini-batch gradient descent on strongly convex smooth loss function.

**Lemma D.3** (LSI constant sequence in Algorithm 1 for strongly convex smooth loss). *Suppose the loss function $\ell(\theta; \mathbf{x})$ in Algorithm 1 is $\lambda$-strongly convex and $\beta$-smooth. If the step-size $\eta < \frac{2}{\lambda+\beta}$, then for any $k = 0, \cdots, K-1$ and $j = 0, \cdots, n/b-1$, the distribution of parameters $\theta_k^j$ in Algorithm 1 satisfies $c_k^j$-log-Sobolev inequality with*

$$c_k^j = \frac{1}{2\eta\sigma^2} \cdot \frac{1}{\sum_{s=0}^{k \cdot n/b+j-1}(1-\eta\lambda)^{2s}} \tag{76}$$

*Proof.* The mini-batch noisy gradient descent update could be written as $\theta_k^{j+1} = f(\theta_k^j) + \mathcal{N}(0, 2\eta\sigma^2\mathbb{I}_d)$, where $f$ is a deterministic mapping on $\mathbb{R}^d$ written as $f(\theta) = \theta - \eta \cdot \frac{\sum_{i \in B} \ell(\theta; \mathbf{x}_i)}{b}$, and $B$ is a mini-batch of size $b$ consisting of indices selected from $0, \cdots, n$.

Because the initialization is point distribution around $\theta_0$, therefore $\theta_0^0$ satisfies log-Sobolev inequality with constant $c_0 = \infty$.

By $\lambda$-strong convexity and $\beta$-smoothness of $\ell(\theta; \mathbf{x})$, and by $\eta < \frac{2}{\lambda+\beta}$, we prove that the mini-batch gradient mapping $f(\theta)$ is $1 - \eta\lambda$-Lipschitz. Further using LSI under Lipchitz mapping (Lemma 16 in Vempala and Wibisono [35]) and under Gaussian convolution (Lemma 17 in Vempala and Wibisono [35]), we prove that

$$\frac{1}{c_k^j} = \frac{(1-\eta\lambda)^2}{c_k^{j-1}} + 2\eta\sigma^2 \tag{77}$$

$$= \frac{(1-\eta\lambda)^{2(k \cdot n/b+j)}}{c_0^0} + 2\eta\sigma^2 \cdot \sum_{s=0}^{k \cdot n/b+j-1}(1-\eta\lambda)^{2s} \tag{78}$$

$$\text{by } c_0^0 = +\infty, \quad = 2\eta\sigma^2 \cdot \sum_{s=0}^{k \cdot n/b+j-1}(1-\eta\lambda)^{2s} \tag{79}$$

This suffices to prove the LSI sequence in Equation (76). □

### D.4 Equivalence Between Lemma 3.2 (Ours) And The Bound in Feldman et al. [18]

We now plug the LSI constant sequence derived in Lemma D.2 into the recursive privacy bound Lemma 3.2, and prove the following privacy dynamics theorems for noisy mini-batch gradient descent under smooth convex loss functions.

**Theorem D.4** (Privacy dynamics under convex smooth loss). *Under fixed mini-batches $B^0, \cdots, B^{n/b-1}$, if the loss function $\ell(\theta; x)$ is convex, $\beta$-smooth, and if its gradient has finite $\ell_2$-sensitivity $S_g$, then Algorithm 1 with step-size $\eta < \frac{2}{\beta}$ satisfies $(\alpha, \varepsilon)$-Rényi DP for data points in the batch $B^{j_0}$, with*

$$\varepsilon_j \leq \frac{\alpha\eta S_g^2}{4\sigma^2 b^2} \cdot \frac{b}{n} \cdot (K-1) + \frac{\alpha\eta S_g^2}{4\sigma^2 b^2} \cdot \frac{1}{n/b-j_0} \tag{80}$$

$$\tag{81}$$

*Proof.* We first observe that for any batch index $j_0 = 0, \cdots, n/b-1$, the privacy bound for data points in batch $B^{j_0}$ in Algorithm 1 (that has $K$ epochs) is equivalent to the privacy bound for data points in the batch $B^0$ after $K-1$ epochs and $n/b-j_0$ iterations in Algorithm 1. That is, if $R_\alpha(\theta_k^j \| \theta'^j_k) \leq \varepsilon_k^j(\alpha)$ is an upper bound for the Rényi divergence between distributions of parameters $\theta_k^j$ and $\theta'^j_k$ in running Algorithm 1 on neighboring datasets $D$ and $D'$, when the differing data point between $D$ and $D'$ is contained in the mini-batch $B^0$, then the privacy bound $\varepsilon$ for data points in the mini-batch $B^{j_0}$ of Algorithm 1 satisfies

$$\varepsilon \leq \varepsilon_{K-1}^{n/b-j_0}(\alpha) \leq \varepsilon_{K-1}^0(\alpha) + \varepsilon_0^{n/b-j_0}(\alpha) \tag{82}$$

$$\leq (K-1) \cdot \varepsilon_0^{n/b}(\alpha) + \varepsilon_0^{n/b-j_0}(\alpha), \tag{83}$$

where the last two inequalities are by composition of Rényi DP guarantees.

Therefore, in the remaining proof, we only prove upper bounds for the terms $\varepsilon_0^j(\alpha)$ for $j = 1, \cdots, n/b$, that are required by Equation (83) for bounding $\varepsilon$.

By Lemma D.2, for any $k = 0, \cdots, K - 1$ and $j = 0, \cdots, n/b - 1$, the distribution of parameters $\theta_k^j$ satisfies log-Sobolev inequality with the following constant $c_k^j$.

$$c_k^j = \frac{1}{2\eta\sigma^2 \cdot (k \cdot n/b + j)} \tag{84}$$

By plugging the LSI constant sequence $\{c_k^j\}$ into Lemma 3.2, and by the noisy mini-batch gradient descent mapping, under convex smooth loss with stepsize $\eta < \frac{2}{\beta}$, is 1-Lipschitz, we prove the following recursive bound for $\varepsilon_k^j(\alpha)$, i.e. the Rényi DP bound for data points in the mini-batch $B^0$ of Algorithm 1. For any $k = 0, \cdots, K - 1$ and any $j = 0, \cdots, n/b - 1$,

$$\varepsilon_0^0(\alpha) = 0 \tag{85}$$

$$\frac{\varepsilon_k^{j+1}(\alpha)}{\alpha} \leq \begin{cases} \frac{\varepsilon_k^j(\alpha)}{\alpha} + \frac{\eta S_g^2}{4\sigma^2 b^2} & \text{if } j = 0 \\ \frac{\varepsilon_k^j(\alpha')}{\alpha'} \cdot \frac{k \cdot n/b + j}{k \cdot n/b + j + 1} & \text{if } j = 1, \cdots, n/b - 1 \end{cases} \tag{86}$$

$$\text{where } \alpha' = (\alpha - 1) \cdot \left(1 + \frac{c \cdot 2\eta\sigma^2}{L^2}\right)^{-1} + 1 \tag{87}$$

$$\varepsilon_{k+1}^0(\alpha) = \varepsilon_k^{n/b}(\alpha) \tag{88}$$

By solving Equation (86) under $k = 0$, we prove that for any $j = 1, \cdots, n/b$, and for any $\alpha > 1$,

$$\frac{\varepsilon_0^j(\alpha)}{\alpha} \leq \frac{\eta S_g^2}{4\sigma^2 b^2} \cdot \prod_{j'=1}^{j-1} \frac{j'}{j' + 1} \tag{89}$$

$$= \frac{\eta S_g^2}{4\sigma^2 b^2} \cdot \frac{1}{j} \tag{90}$$

By plugging Equation (90) into Equation (83), we prove the privacy bound Equation (80) in the theorem statement. □

Theorem D.4 is equivalent to Theorem 23 in Feldman et al. [18] for Algorithm 1 under single-epoch setting (where $K = 1$) with batch-size $b = 1$. For multi-epoch setting with $K = n$ and batch-size $b = 1$, Theorem D.4 is equivalent to Theorem 35 in Feldman et al. [18].

### D.5   Proof for Theorem 3.3

We now plug the LSI constant sequence derived in Lemma D.3 into the recursive privacy bound Lemma 3.2, and prove the following privacy dynamics theorems for noisy mini-batch gradient descent under strongly convex smooth loss functions.

*Theorem 3.3.* Conditioned on a fixed sequence of partitioned mini-batches $B^0, \cdots, B^{n/b-1}$ in Line 3, if the loss function is $\lambda$-strongly convex, $\beta$-smooth and its gradient has $\ell_2$-sensitivity $S_g$, then running Algorithm 1 for $K \geq 1$ epochs with step-size $\eta < \frac{2}{\lambda+\beta}$, satisfies $(\alpha, \varepsilon)$-Rényi DP for data points in the batch $B^{j_0}$, with

$$\varepsilon \leq \varepsilon_0^{\lfloor \frac{n}{2b} \rfloor}(\alpha) \cdot \frac{1 - (1 - \eta\lambda)^{2 \cdot (K-1) \cdot (n/b - \lfloor \frac{n}{2b} \rfloor)}}{1 - (1 - \eta\lambda)^{2 \cdot (n/b - \lfloor \frac{n}{2b} \rfloor)}} + \varepsilon_0^{n/b-j_0}(\alpha) \tag{91}$$

where $\varepsilon_0^j(\alpha) = \frac{\alpha \eta S_g^2}{4\sigma^2 b^2} \cdot (1 - \eta\lambda)^{2 \cdot (j-1)} \cdot \frac{1}{\sum_{s=0}^{j-1}(1-\eta\lambda)^{2s}}$ for any $j = 1, \cdots, \frac{n}{b}$ (we assume $\frac{n}{b} \geq 2$).

*Proof.* We first observe that for any batch index $j_0 = 0, \cdots, n/b - 1$, the privacy bound for data points in batch $B^{j_0}$ in Algorithm 1 (that has $K$ epochs) is equivalent to the privacy bound for data

points in the batch $B^0$ after $K-1$ epochs and $n/b - j_0$ iterations in Algorithm 1. That is, if $R_\alpha(\theta_k^j \| \theta'^j_k) \le \varepsilon_k^j(\alpha)$ is an upper bound for the Rényi divergence between distributions of parameters $\theta_k^j$ and $\theta'^j_k$ in running Algorithm 1 on neighboring datasets $D$ and $D'$, when the differing data point between $D$ and $D'$ is contained in the mini-batch $B^0$, then the privacy bound $\varepsilon$, for running Algorithm 1 on neighboring datasets $D$ and $D'$ that differs in a point in the mini-batch $B^{j_0}$, satisfies

$$\varepsilon \le \varepsilon_{K-1}^{n/b-j_0}(\alpha) \le \varepsilon_{K-1}^0(\alpha) + \varepsilon_0^{n/b-j_0}(\alpha), \tag{92}$$

where the last inequality is by composition of Rényi DP guarantees.

Therefore, in the remaining proof, we only prove upper bounds for the terms $\varepsilon_0^{n/b-1-j_0}(\alpha)$ and $\varepsilon_{K-1}^0(\alpha)$, that are required by Equation (92) for bounding $\varepsilon$.

By Lemma D.3, for any $k = 0, \cdots, K-1$ and $j = 0, \cdots, n/b-1$, the distribution of parameters $\theta_k^j$ satisfies log-Sobolev inequality with the following constant $c_k^j$.

$$c_k^j = \frac{1}{2\eta\sigma^2} \cdot \frac{1}{\sum_{s=0}^{k \cdot n/b + j - 1}(1 - \eta\lambda)^{2s}} \tag{93}$$

By plugging the LSI constant sequence $\{c_k^j\}$ proved in Lemma D.3 into the proved recursive privacy bound Lemma 3.2, we prove that for running noisy mini-batch gradient descent on neighboring datasets that differ in a data point in the first mini-batch $B^0$, the following recursion for the Rényi divergence bound $\varepsilon_k^j(\alpha)$ holds: if the loss function is $\lambda$-strongly convex and $\beta$-smooth, and if the stepsize $\eta < \frac{2}{\lambda+\beta}$, then for any $k = 0, \cdots, K-1$, any $j = 0, \cdots, n/b-1$, and any $\alpha > 1$,

$$\varepsilon_0^0(\alpha) = 0 \tag{94}$$

$$\frac{\varepsilon_k^{j+1}(\alpha)}{\alpha} \le \begin{cases} \frac{\varepsilon_k^j(\alpha)}{\alpha} + \frac{\eta S_g^2}{4\sigma^2 b^2} & \text{if } j = 0 \\ \frac{\varepsilon_k^j(\alpha')}{\alpha'} \cdot (1-\eta\lambda)^2 \cdot \frac{\sum_{s=0}^{k \cdot n/b + j - 1}(1-\eta\lambda)^{2s}}{\sum_{s=0}^{k \cdot n/b + j}(1-\eta\lambda)^{2s}} & \text{if } j = 1, \cdots, n/b-1 \end{cases} \tag{95}$$

where $\alpha' = (\alpha - 1) \cdot \left(1 + \frac{c \cdot 2\eta\sigma^2}{L^2}\right)^{-1} + 1$ \hfill (96)

$$\varepsilon_{k+1}^0(\alpha) = \varepsilon_k^{n/b}(\alpha) \tag{97}$$

We now solve the above recursion, thus bounding the privacy loss $\varepsilon$ of Algorithm 1 in Equation (92).

1. We first prove a bound for the term $\varepsilon_0^{n/b-j_0}(\alpha)$ in Equation (92). By solving Equation (95) under $k = 0$, we prove that for any $j = 1, \cdots, n/b$, and for any $\alpha > 1$,

$$\frac{\varepsilon_0^j(\alpha)}{\alpha} \le \frac{\eta S_g^2}{4\sigma^2 b^2} \cdot \prod_{j'=1}^{j-1}\left((1-\eta\lambda)^2 \cdot \frac{\sum_{s=0}^{j'-1}(1-\eta\lambda)^{2s}}{\sum_{s=0}^{j'}(1-\eta\lambda)^{2s}}\right) \tag{98}$$

$$= \frac{\eta S_g^2}{4\sigma^2 b^2} \cdot (1-\eta\lambda)^{2 \cdot (j-1)} \cdot \frac{1}{\sum_{s=0}^{j-1}(1-\eta\lambda)^{2s}} \tag{99}$$

2. We now prove a bound for the term $\varepsilon_{K-1}^0(\alpha)$ in Equation (92).

    The term $\varepsilon_k^j(\alpha)$ corresponds to the privacy bound for the data points in the batch $B^0$ for running Algorithm 1 with $k$ epochs plus $j$ iteration. This is equivalent to the composition $\mathcal{A}_2 \circ \mathcal{A}_1$ of two sub-mechanisms $\mathcal{A}_1$ and $\mathcal{A}_2$, where $\mathcal{A}_1$ corresponds to running Algorithm 1 with $K = k$ epochs, and $\mathcal{A}_2$ corresponds to running Algorithm 1 with only $j$ iterations. Therefore, by the composition theorem for Rényi DP guarantees [28], we prove the following alternative recursive privacy bound for $\varepsilon_k^j(\alpha)$. For any $k = 0, \cdots, K-1$ and any $j = 0, \cdots, n/b-1$,

$$\varepsilon_k^{j+1}(\alpha) \le \varepsilon_k^0(\alpha) + \varepsilon_0^{j+1}(\alpha) \tag{100}$$

We now prove an upper bound for $\varepsilon_k^j(\alpha)$, by carefully combining the original recursion Equation (95) and the new alternative recursion Equation (100) (obtained by composition theorem). We use the original recursion Equation (100) for recursively bounding $\varepsilon_k^{j+1}(\alpha)$ during the first half of one epoch, i.e., for $j = 0, \cdots, \lfloor \frac{n}{2b} \rfloor - 1$, and then we use the new alternative recursive bound Equation (95) for the second half of one epoch, i.e., for $j = \lfloor \frac{n}{2b} \rfloor, \cdots, n/b - 1$. [5] Via this combination, we obtain a new combined recursion for privacy bound as follows. For any $k = 0, \cdots, K - 1$,

$$\frac{\varepsilon_k^{j+1}(\alpha)}{\alpha} \leq \begin{cases} \frac{\varepsilon_k^0(\alpha) + \varepsilon_0^{j+1}(\alpha)}{\alpha} & \text{if } j = 0, \cdots, \lfloor \frac{n}{2b} \rfloor - 1 \\ \frac{\varepsilon_k^j(\alpha')}{\alpha'} \cdot (1 - \eta\lambda)^2 \cdot \frac{\sum_{s=0}^{k \cdot n/b + j - 1}(1 - \eta\lambda)^{2s}}{\sum_{s=0}^{k \cdot n/b + j}(1 - \eta\lambda)^{2s}} & \text{if } j = \lfloor \frac{n}{2b} \rfloor, \cdots, n/b - 1 \end{cases}$$

(101)

where $\alpha' = (\alpha - 1) \cdot \left(1 + \frac{c \cdot 2\eta\sigma^2}{L^2}\right)^{-1} + 1$.

We now solve this new recursion Equation (101) for one epoch, by accumulating $j = 0, \cdots, n/b - 1$. We prove that for any $k = 0, \cdots, K - 1$,

$$\frac{\varepsilon_{k+1}^0(\alpha)}{\alpha} = \frac{\varepsilon_k^{n/b}(\alpha)}{\alpha}$$

(102)

$$\leq \frac{\varepsilon_k^0(\tilde{\alpha}) + \varepsilon_0^{\lfloor \frac{n}{2b} \rfloor}(\tilde{\alpha})}{\tilde{\alpha}} \cdot \prod_{j'=\lfloor \frac{n}{2b} \rfloor}^{n/b - 1}(1 - \eta\lambda)^2 \cdot \frac{\sum_{s=0}^{k \cdot n/b + j - 1}(1 - \eta\lambda)^{2s}}{\sum_{s=0}^{k \cdot n/b + j}(1 - \eta\lambda)^{2s}}$$

(103)

$$\leq \frac{\left(\varepsilon_k^0(\tilde{\alpha}) + \varepsilon_0^{\lfloor \frac{n}{2b} \rfloor}(\tilde{\alpha})\right)}{\tilde{\alpha}} \cdot (1 - \eta\lambda)^{2 \cdot (n/b - \lfloor \frac{n}{2b} \rfloor)}$$

(104)

where the RDP order $\tilde{\alpha} > 1$ is the $n/b - \lfloor \frac{n}{2b} \rfloor$ fold mapped value of $\alpha$ under repeated mappings $\alpha \leftarrow (\alpha - 1) \cdot \left(1 + \frac{c \cdot 2\eta\sigma^2}{L^2}\right)^{-1} + 1$.

We now further solve this new recursion Equation (101) for multiple epochs, by accumulating Equation (104) for $k = 0, 1, \cdots, K - 1$. We prove that for any $k = 0, 1, \cdots, K - 1$,

$$\frac{\varepsilon_{k+1}^0(\alpha)}{\alpha} \leq \frac{\varepsilon_0^0(\tilde{\alpha})}{\tilde{\alpha}} \cdot (1 - \eta\lambda)^{2 \cdot (k+1) \cdot (n/b - \lfloor \frac{n}{2b} \rfloor)} + \frac{\varepsilon_0^{\lfloor \frac{n}{2b} \rfloor}(\tilde{\alpha})}{\tilde{\alpha}} \cdot \sum_{k'=1}^{k+1}(1 - \eta\lambda)^{2 \cdot k' \cdot (n/b - \lfloor \frac{n}{2b} \rfloor)}$$

(105)

for some RDP order $\tilde{\alpha} > 1$ that is the $(k + 1) \cdot (n/b - \lfloor \frac{n}{2b} \rfloor)$ fold mapped value of $\alpha$ under repeated mapping $\alpha \leftarrow (\alpha - 1) \cdot \left(1 + \frac{c \cdot 2\eta\sigma^2}{L^2}\right)^{-1} + 1$.

By further substituting $\varepsilon_0^0(\tilde{\alpha}) = 0$ at the initialization point $\theta_0$ for any $\tilde{\alpha} > 1$ in the above equation, we prove that for any $k = 0, \cdots, K - 1$

$$\frac{\varepsilon_{k+1}^0(\alpha)}{\alpha} = \frac{\varepsilon_0^{\lfloor \frac{n}{2b} \rfloor}(\tilde{\alpha})}{\tilde{\alpha}} \cdot (1 - \eta\lambda)^{2 \cdot (n/b - \lfloor \frac{n}{2b} \rfloor)} \cdot \frac{1 - (1 - \eta\lambda)^{2 \cdot (k+1) \cdot (n/b - \lfloor \frac{n}{2b} \rfloor)}}{1 - (1 - \eta\lambda)^{2 \cdot (n/b - \lfloor \frac{n}{2b} \rfloor)}}$$

(106)

for some RDP order $\tilde{\alpha} > 1$.

By setting $k = K - 2$ in Equation (106), we prove that for $K \geq 2$

$$\frac{\varepsilon_{K-1}^0(\alpha)}{\alpha} \leq \frac{\varepsilon_0^{\lfloor \frac{n}{2b} \rfloor}(\tilde{\alpha})}{\tilde{\alpha}} \cdot \frac{1 - (1 - \eta\lambda)^{2 \cdot (K-1) \cdot (n/b - \lfloor \frac{n}{2b} \rfloor)}}{1 - (1 - \eta\lambda)^{2 \cdot (n/b - \lfloor \frac{n}{2b} \rfloor)}}$$

(107)

for some RDP order $\tilde{\alpha} > 1$. By the format of $\varepsilon_0^j(\alpha)$ for any $\alpha > 1$ in our proof Equation (99) (i.e., $\frac{\varepsilon_0^j(\alpha)}{\alpha}$ is bounded by a constant for all $\alpha > 1$), we prove that Equation (107) is equivalent

---

[5]This seemingly artificial way of separating one epoch into two halves, and using the recursive bounds separately in each half, is for obtaining a small privacy bound at convergence.

to the following equation

$$\varepsilon_{K-1}^0(\alpha) \leq \varepsilon_0^{\lfloor \frac{n}{2b} \rfloor}(\alpha) \cdot \frac{1 - (1-\eta\lambda)^{2 \cdot (K-1) \cdot (n/b - \lfloor \frac{n}{2b} \rfloor)}}{1 - (1-\eta\lambda)^{2 \cdot (n/b - \lfloor \frac{n}{2b} \rfloor)}} \tag{108}$$

For $K = 1$, by definition, we compute that $\varepsilon_{K-1}^0(\alpha) = 0 =$Right hand side of Equation (108), therefore Equation (108) also holds for $K = 1$.

We now plug our bound Equation (108) into Equation (92), and prove a bound for $\varepsilon$ as follows.

$$\varepsilon \leq \varepsilon_{K-1}^0(\alpha) + \varepsilon_0^{n/b - j_0}(\alpha) \tag{109}$$

$$\leq \varepsilon_0^{\lfloor \frac{n}{2b} \rfloor}(\alpha) \cdot \frac{1 - (1-\eta\lambda)^{2 \cdot (K-1) \cdot (n/b - \lfloor \frac{n}{2b} \rfloor)}}{1 - (1-\eta\lambda)^{2 \cdot (n/b - \lfloor \frac{n}{2b} \rfloor)}} + \varepsilon_0^{n/b - j_0}(\alpha) \tag{110}$$

where the term $\varepsilon_0^j(\alpha)$ is upper bounded by Equation (99). This gives the privacy bound Equation (3) in the theorem statement. $\qquad\square$

### D.6 Explanations for the privacy bound derived from Balle et al. [5] in Figure 1

For convenience, we first translate [5, Theorem 5] into the symbols used in this paper, as well as under mini-batch with size $b > 1$, as follows.

**Theorem D.5** (Balle et al. [5]). *Let $\ell(\theta, \mathbf{x})$ be an L-Lipschitz, $\beta$-smooth, $\lambda$-strongly convex loss function. If $\eta \leq \frac{2}{\beta+\lambda}$, then conditioned on a fixed sequence of partitioned mini-batches $B^0, \cdots, B^{n/b-1}$ in Line 3, Algorithm 1 satisfies $(\alpha, \varepsilon)$-Rényi DP for data points in the mini-batch $j_0$, with*

$$\varepsilon = \begin{cases} \alpha \cdot \frac{2(L/b)^2}{2\eta\sigma^2} & j_0 = n/b - 1 \\ \alpha \cdot \frac{2(L/b)^2}{(n/b - j_0 - 1) \cdot 2\eta\sigma^2} \left(1 - \frac{2\eta\beta\lambda}{\beta+\lambda}\right)^{\frac{n/b - j_0}{2}} & j_0 = 0, \cdots, n/b - 2 \end{cases} \tag{111}$$

The variables from Balle et al. [5] that we replaced are as follows: $L \to L/b$, this is because, in Algorithm 1, we additionally average each per-example gradient with the mini-batch size $b$; $i \to j_0 + 1$, this is because the index used in our paper starts from $j_0 = 0$, while the index used in Balle et al. [5] starts from $i = 1$; $n \to n/b$, this is because under mini-batch size $b > 1$, the number of iterations in one epoch in our paper is $n/b$, while the number of iteration in Balle et al. [5] equals the size of the datasets $n$; $\sigma^2 \to 2\eta\sigma^2$, this is because in our paper, we follow the noise scaling with variance $2\eta\sigma^2$ that is related to stepsize $\eta$, as in SGLD, while Balle et al. [5] use the standard Gaussian noise with variance $\sigma^2$.

By further substituting the Lipschitz requirement with its equivalent sensitivity assumption, i.e. $L \to \eta S_g/2$, and by using Rényi DP composition [28] over the epochs, we obtain that the privacy bound in Balle et al. [5, Theorem 5] is $\frac{\alpha \cdot \eta S_g^2}{4 \cdot (n/b-1) \cdot b^2\sigma^2} \cdot \left(1 - \frac{2\eta\beta\lambda}{\beta+\lambda}\right)^{\frac{n}{2b}} \cdot K$ when the differing data point is in the first batch, and the bound is $\frac{\alpha \cdot \eta S_g^2}{4 \cdot (n/b-1) \cdot b^2\sigma^2} \cdot \left(1 - \frac{2\eta\beta\lambda}{\beta+\lambda}\right)^{\frac{n}{2b}} \cdot (K-1) + \frac{\alpha \cdot \eta S_g^2}{4b^2\sigma^2}$ when the differing data point is in the last batch.

### D.7 Revisiting noisy GD: A tighter bound than [15, Corollary 1]

In this section, we prove a new converging hidden-state privacy bound for the noisy GD algorithm, which is a special case of Algorithm 1 under $b = n$. We then show that our bound is slightly tighter than [15, Corollary 1] and also admits a conceptually simpler proof (our proof relies on the privacy amplification by randomized post-processing results in Section 3). We first state our privacy bound and its proof as follows.

**Theorem D.6.** *Let $\ell(\theta; \mathbf{x})$ be a $\lambda$-strongly convex, and $\beta$-smoooth loss function, with a finite total gradient sensitivity $S_g$, then the noisy gradient descent algorithm with step-size $\eta < \frac{1}{\beta}$, satisfies $(\alpha, \varepsilon)$ Rényi Differential Privacy with*

$$\varepsilon \leq \frac{\alpha\eta S_g^2}{2\sigma^2 n^2} \cdot \sum_{k=1}^K \left(1 - \frac{\eta\lambda}{2}\right)^k. \tag{112}$$

*Proof.* We first offer a new perspective of viewing a noisy GD update as follows. Recall that one noisy GD update in Algorithm 1 is written as

$$\theta_{k+1}^0 = \theta_k^0 - \eta \cdot g(\theta_k^0; D) + \sqrt{2\eta\sigma^2} \cdot \mathcal{N}(0, \mathbb{I}_d) \text{ where } g(\theta_k^j; D) = \frac{1}{n}\sum_{i=1}^n \nabla\ell(\theta_k^j; \mathbf{x}_i) \quad (113)$$

Therefore, this update is equivalent to the following two steps:

$$\theta_k^{\frac{1}{2}} = \theta_k^0 - \eta \cdot g(\theta_k^0; D) + \sqrt{\eta\sigma^2} \cdot \mathcal{N}(0, \mathbb{I}_d) \text{ where } g(\theta_k^0; D) = \frac{1}{n}\sum_{i=1}^n \nabla\ell(\theta_k^0; \mathbf{x}_i) \quad (114)$$

$$\theta_{k+1}^0 = \theta_k^{\frac{1}{2}} + \sqrt{\eta\sigma^2} \cdot \mathcal{N}(0, \mathbb{I}_d) \quad (115)$$

That is, we view each noisy GD update as another noisy GD update with smaller noise scale followed by pure additive Gaussian noise.

Moreover, by Lemma D.3, we prove that distributions of $\theta_k^0$ satisfies $c_k^0$-log Sobolev inequality with

$$c_k^0 = \frac{1}{2\eta\sigma^2} \cdot \frac{1}{\sum_{s=0}^{k-1}(1-\eta\lambda)^{2s}} \quad (116)$$

Therefore, by further using LSI under Lipschitz mapping ([35, Lemma 16]) and under Gaussian convolution ([35, Lemma 17]), we prove that the distribution of $\theta_k^{\frac{1}{2}}$ satisfies $c_k^{\frac{1}{2}}$-log Sobolev inequality with

$$\frac{1}{c_k^{\frac{1}{2}}} = \frac{(1-\eta\lambda)^2}{c_k^0} + \eta\sigma^2 \quad (117)$$

$$= 2\eta\sigma^2 \cdot \sum_{s=1}^k (1-\eta\lambda)^{2s} + \eta\sigma^2 \quad (118)$$

Therefore, by applying composition on the conceptual step from $\theta_k^0 \to \theta_k^{\frac{1}{2}}$, we prove that

$$\frac{R_\alpha(\theta_k^{\frac{1}{2}}\|\theta_k'^{\frac{1}{2}})}{\alpha} \leq \frac{R_\alpha(\theta_k^0\|\theta_k'^0)}{\alpha} + \frac{\eta S_g^2}{2\sigma^2 n^2} \quad (119)$$

For the remaining conceptual step from $\theta_k^{\frac{1}{2}} \to \theta_{k+1}^0$, we note that pure additive Gaussian noise is equivalent to identity mapping convolved with Gaussian noise. Therefore, we apply $i_0 \notin B_k^j$ case of Lemma 3.2 with $L = 1$ (for identity mapping) and noise standard deviation $\frac{\sigma}{\sqrt{2}}$, and prove that

$$\frac{R_\alpha(\theta_{k+1}^0\|\theta_{k+1}'^0)}{\alpha} \leq \frac{R_{\alpha'}(\theta_k^{\frac{1}{2}}\|\theta_k'^{\frac{1}{2}})}{\alpha'} \cdot \left(1 + c_k^{\frac{1}{2}} \cdot \eta\sigma^2\right)^{-1} \text{ with } \alpha' = \frac{\alpha-1}{1 + c_k^{\frac{1}{2}} \cdot \eta\sigma^2} + 1 \quad (120)$$

Therefore, by combining (119) and (120), we prove the following recursive Rényi DP bound.

$$\frac{R_\alpha(\theta_{k+1}^0\|\theta_{k+1}'^0)}{\alpha} \leq \left(\frac{R_{\alpha'}(\theta_k^0\|\theta_k'^0)}{\alpha'} + \frac{\eta S_g^2}{2\sigma^2 n^2}\right) \cdot \left(1 + c_k^{\frac{1}{2}} \cdot \eta\sigma^2\right)^{-1} \text{ with } \alpha' = \frac{\alpha-1}{1 + c_k^{\frac{1}{2}} \cdot \eta\sigma^2} + 1$$
$$(121)$$

By further plugging in the LSI constant sequence (117), we simplify Equation (121) into the following inequality.

$$\frac{R_\alpha(\theta^0_{k+1}\|\theta'^0_{k+1})}{\alpha} \leq \left(\frac{R_{\alpha'}(\theta^0_k\|\theta'^0_k)}{\alpha'} + \frac{\eta S_g^2}{2\sigma^2 n^2}\right) \cdot \left(1 - \frac{1}{2\sum_{s=0}^k (1-\eta\lambda)^{2s}}\right) \tag{122}$$

$$\leq \left(\frac{R_{\alpha'}(\theta^0_k\|\theta'^0_k)}{\alpha'} + \frac{\eta S_g^2}{2\sigma^2 n^2}\right) \cdot \left(1 - \frac{1}{2\sum_{s=0}^{+\infty} (1-\eta\lambda)^{2s}}\right) \tag{123}$$

$$= \left(\frac{R_{\alpha'}(\theta^0_k\|\theta'^0_k)}{\alpha'} + \frac{\eta S_g^2}{2\sigma^2 n^2}\right) \cdot \left(1 - \frac{1-(1-\eta\lambda)^2}{2}\right) \tag{124}$$

$$\leq \left(\frac{R_{\alpha'}(\theta^0_k\|\theta'^0_k)}{\alpha'} + \frac{\eta S_g^2}{2\sigma^2 n^2}\right) \cdot \left(1 - \frac{\eta\lambda}{2}\right), \tag{125}$$

where $\alpha' = (\alpha - 1) \cdot \left(1 - \frac{1}{2\sum_{s=0}^k (1-\eta\lambda)^{2s}}\right) + 1$, and the last inequality Equation (125) is because of the inequality $\eta\lambda < \frac{\lambda}{\beta} \leq 1$ that is ensured by the condition $\eta < \frac{1}{\beta}$.

Therefore, by solving the recursion Equation (125) from $k = 0, \cdots, K-1$, and by using $R_\alpha(\theta^0_0\|\theta'^0_0) = 0$ for any $\alpha > 1$, we prove the theorem statement in Theorem D.6.

$$\frac{R_\alpha(\theta^0_K\|\theta'^0_K)}{\alpha} \leq \frac{\eta S_g^2}{2\sigma^2 n^2} \cdot \sum_{k=1}^K \left(1 - \frac{\eta\lambda}{2}\right)^k \tag{126}$$

$\square$

**Comparison with [15, Corollary 1]**  We now compare our privacy bound Theorem D.6 with [15, Corollary 1] and show that our bound is tighter. For convenience, we now repeat the privacy bound for noisy gradient descent (i.e. when $b = n$ in Algorithm 1) that is proved in [15, Corollary 1] below.

**Theorem D.7** (Corollary 1 in Chourasia et al. [15]). *Let $\ell(\theta; \mathbf{x})$ be a $\lambda$-strongly convex, and $\beta$-smoooth loss function, with a finite total gradient sensitivity $S_g$, then the noisy gradient descent algorithm with start parameter $\theta_0 \sim \mathcal{N}(0, \frac{2\sigma^2}{\lambda}\mathcal{I}_d)$, and step-size $\eta < \frac{1}{\beta}$, satisfies $(\alpha, \varepsilon)$ Rényi Differential Privacy with*

$$\varepsilon \leq \frac{\alpha S_g^2}{\lambda\sigma^2 n^2}(1 - e^{-\lambda\eta K/2}). \tag{127}$$

Because $1 - x < e^{-x}$ for $x \neq 0$, we are able to further relax the bound in Theorem D.6 as follows.

$$\varepsilon \leq \frac{\alpha\eta S_g^2}{2\sigma^2 n^2} \cdot \sum_{k=1}^K \left(1 - \frac{\eta\lambda}{2}\right)^k \tag{128}$$

$$< \frac{\alpha\eta S_g^2}{2\sigma^2 n^2} \cdot \left(1 - \frac{\eta\lambda}{2}\right) \cdot \sum_{k=0}^{K-1} e^{-\frac{\eta\lambda k}{2}} \tag{129}$$

$$= \frac{\alpha\eta S_g^2}{2\sigma^2 n^2} \cdot \left(1 - \frac{\eta\lambda}{2}\right) \cdot \frac{1 - e^{-\lambda\eta K}}{1 - e^{-\eta\lambda/2}} \tag{130}$$

Because $1 - e^{-x} > x \cdot e^{-x}$ for $x > 0$, we prove that $1 - e^{-\eta\lambda/2} \geq \frac{\lambda\eta}{2} \cdot e^{-\frac{\lambda\eta}{2}}$. By plugging this inequality to Equation (130), we prove that

$$\varepsilon < \frac{\alpha S_g^2}{\lambda\sigma^2 n^2} \cdot \frac{1 - \frac{\eta\lambda}{2}}{e^{-\frac{\lambda\eta}{2}}} \cdot (1 - e^{-\lambda\eta K}) \tag{131}$$

By again using $1 - x < e^{-x}$ for $x \neq 0$ into Equation (131), we prove that

$$\varepsilon < \frac{\alpha S_g^2}{\lambda \sigma^2 n^2} \cdot (1 - e^{-\lambda \eta K}) = \text{RHS of [15, Corollary 1]} \tag{132}$$

This shows that the privacy bound Theorem D.6 enabled by our analysis is strictly tighter than [15, Corollary 1].

## E   Proof for Section 4

*Lemma 4.1.* Let $\mu_1, \cdots, \mu_m$ and $\nu_1, \cdots, \nu_m$ be measures over $R^d$. Then for any $\alpha \geq 1$, and any $p_1, \cdots, p_m \geq 0$ that satisfies $p_1 + \cdots + p_m = 1$,

$$e^{(\alpha-1) \cdot R_\alpha(\sum_{j=1}^m p_j \mu_j \| \sum_{j=1}^m p_j \nu_j)} \leq \sum_{j=1}^m p_j \cdot e^{(\alpha-1) \cdot R_\alpha(\mu_j \| \nu_j)} \tag{133}$$

*Proof.* By definition of Rényi divergence,

$$e^{(\alpha-1) \cdot R_\alpha(\mu \| \nu)} = \int \left( \frac{\mu(\theta)}{\nu(\theta)} \right)^\alpha \cdot \mu(\theta) d\theta \tag{134}$$

Therefore by definition of $f$-divergence, $e^{(\alpha-1) \cdot R_\alpha(\mu \| \nu)}$ is $f$-divergence with $f(x) = x^\alpha$. By the convexity of $f$ when $\alpha \geq 1$, and by applying Theorem 3.1 in Taneja and Kumar [33], we prove that the f-divergence $e^{(\alpha-1) \cdot R_\alpha(\mu \| \nu)}$ is jointly convex in arguments $\mu, \nu$. □

### E.1   Proof for Theorem 4.2

*Theorem 4.2* (Privacy dynamics for "shuffle and partition" mini-batch gradient descent). If the loss function $\ell(\theta; x)$ is $\lambda$-strongly convex, $\beta$-smooth, and if its gradient has finite $\ell_2$-sensitivity $S_g$, then for $K \geq 1$ and $\frac{n}{b} \geq 2$, Algorithm 1 with stepsize $\eta < \frac{2}{\lambda + \eta}$ satisfies $(\alpha, \varepsilon)$-Rényi DP for all data points with

$$\varepsilon \leq \varepsilon_0^{\lfloor \frac{n}{2b} \rfloor}(\alpha) \cdot \frac{1 - (1-\eta\lambda)^{2 \cdot (K-1) \cdot (n/b - \lfloor \frac{n}{2b} \rfloor)}}{1 - (1-\eta\lambda)^{2 \cdot (n/b - \lfloor \frac{n}{2b} \rfloor)}} + \frac{1}{\alpha - 1} \cdot \log \left( \underset{0 \leq j_0 < n/b}{Avg} \ e^{(\alpha-1)\varepsilon_0^{n/b-j_0}(\alpha)} \right) \tag{135}$$

where the terms $\varepsilon_0^j(\alpha)$ is upper-bounded for any $j = 1, \cdots, n/b$ as follows.

$$\varepsilon_0^j(\alpha) \leq \frac{\alpha \eta S_g^2}{4\sigma^2 b^2} \cdot (1-\eta\lambda)^{2 \cdot (j-1)} \cdot \frac{1}{\sum_{s=0}^{j-1}(1-\eta\lambda)^{2s}} \tag{136}$$

*Proof.* Our proof relies on the joint convexity of the scaled exponentiated Rényi divergence $e^{(\alpha-1)R_\alpha(\mu \| \nu)}$ in its arguments $\mu$ and $\nu$. By further using the batch decomposition in Section 2, we prove

$$\varepsilon = \frac{1}{\alpha - 1} \log e^{(\alpha-1) \cdot R_\alpha(\theta_K^0 \| \theta'^0_K)} \tag{137}$$

$$\leq \frac{1}{\alpha - 1} \log \left( \sum_{B^0, \cdots, B^{n/b-1} \text{in Line 3}} p(B^0, \cdots, B^{n/b-1}) \cdot e^{(\alpha-1) \cdot R_\alpha \left( p(\theta_K^0 | B^0, \cdots, B^{n/b-1}) \| p(\theta'^0_K | B^0, \cdots, B^{n/b-1}) \right)} \right) \tag{138}$$

Therefore, by plugging the privacy bound of Algorithm 1 proved in Theorem 3.3 into Equation (138), depending on which mini-batch contains the differing data point with any index $i_0$, we further prove

that

$$\varepsilon \le \frac{1}{\alpha - 1} \log \left( \sum_{j_0=0}^{n/b-1} p(i_0 \in B^{j_0}) \cdot e^{(\alpha-1) \cdot \left( \varepsilon_0^{\lfloor \frac{n}{2b} \rfloor}(\alpha) \cdot \frac{1-(1-\eta\lambda)^{2\cdot(K-1)\cdot(n/b-\lfloor \frac{n}{2b} \rfloor)}}{1-(1-\eta\lambda)^{2\cdot(n/b-\lfloor \frac{n}{2b} \rfloor)}} + \varepsilon_0^{n/b-j_0}(\alpha) \right)} \right)$$

(139)

$$= \frac{1}{\alpha - 1} \log \left( \sum_{j_0=0}^{n/b-1} \frac{b}{n} \cdot e^{(\alpha-1) \cdot \left( \varepsilon_0^{\lfloor \frac{n}{2b} \rfloor}(\alpha) \cdot \frac{1-(1-\eta\lambda)^{2\cdot(K-1)\cdot(n/b-\lfloor \frac{n}{2b} \rfloor)}}{1-(1-\eta\lambda)^{2\cdot(n/b-\lfloor \frac{n}{2b} \rfloor)}} + \varepsilon_0^{n/b-j_0}(\alpha) \right)} \right)$$

(140)

$\square$

***Explanations for the terms in Theorem 4.2.*** The privacy bound Theorem 4.2 is strictly smaller than the composition-based privacy bound (derived from DP-SGD[1] and SGM [29] analysis) after $\frac{1}{\lambda\eta} + \frac{n}{b}$ epochs. More specifically, the first term in Equation (12) is strictly smaller than the composition-based privacy bound after $\frac{1}{\lambda\eta}$ epochs, and the second term in Equation (12) is strictly smaller than the composition-based privacy bound after $n/b$ epochs (where $n/b \ge 2$). We explain the terms more specifically as follows.

1. The sensitivity for one mini-batch gradient update in Algorithm 1 is $\eta \cdot \frac{S_g}{b}$, the standard deviation of Gaussian noise added in one update is $\sqrt{2\eta\sigma^2}$, and the sampling probability for each data point is $\frac{b}{n}$. By Abadi et al. [1] and Mironov et al. [29], the composition-based privacy bound for one iteration is larger than $\frac{b^2}{n^2} \frac{\alpha \cdot \eta^2 \cdot S_g^2 / b^2}{2 \cdot 2\eta\sigma^2} = \frac{b^2}{n^2} \frac{\alpha \cdot \eta S_g^2}{4\sigma^2 b^2}$. By Rényi DP composition over $n/b$ iterations, the composition-based privacy bound for one epoch is larger than $\frac{b}{n} \cdot \frac{\alpha \cdot \eta S_g^2}{4\sigma^2 b^2}$.

2. The first term is upper bounded as follows, which is smaller than the composition-based privacy bound after $\frac{1}{\lambda\eta}$ epochs.

$$\varepsilon_0^{\lfloor \frac{n}{2b} \rfloor}(\alpha) \cdot \frac{1-(1-\eta\lambda)^{2\cdot(K-1)\cdot(n/b-\lfloor \frac{n}{2b} \rfloor)}}{1-(1-\eta\lambda)^{2\cdot(n/b-\lfloor \frac{n}{2b} \rfloor)}}$$

(141)

$$\le \frac{\alpha\eta S_g^2}{4\sigma^2 b^2} \cdot (1-\eta\lambda)^{2\cdot(\lfloor \frac{n}{2b} \rfloor - 1)} \cdot \frac{1}{\sum_{s=0}^{\lfloor \frac{n}{2b} \rfloor - 1}(1-\eta\lambda)^{2s}} \cdot \frac{1-(1-\eta\lambda)^{2\cdot(K-1)\cdot(n/b-\lfloor \frac{n}{2b} \rfloor)}}{1-(1-\eta\lambda)^{2\cdot(n/b-\lfloor \frac{n}{2b} \rfloor)}}$$

(142)

$$= \frac{\alpha\eta S_g^2}{4\sigma^2 b^2} \cdot (1-\eta\lambda)^{2\cdot(\lfloor \frac{n}{2b} \rfloor - 1)} \cdot \frac{1-(1-\eta\lambda)^2}{1-(1-\eta\lambda)^{2\cdot\lfloor \frac{n}{2b} \rfloor}} \cdot \frac{1-(1-\eta\lambda)^{2\cdot(K-1)\cdot(n/b-\lfloor \frac{n}{2b} \rfloor)}}{1-(1-\eta\lambda)^{2\cdot(n/b-\lfloor \frac{n}{2b} \rfloor)}}$$

(143)

$$\le \frac{\alpha\eta S_g^2}{4\sigma^2 b^2} \cdot \frac{1}{1-(1-\eta\lambda)^{2\cdot(n/b-\lfloor \frac{n}{2b} \rfloor)}} \quad \text{(by } (1-\eta\lambda)^x \text{ is monotonically decreasing for } x \in \mathbb{R})$$

(144)

$$\le \frac{\alpha\eta S_g^2}{4\sigma^2 b^2} \cdot \frac{1}{2 \cdot (n/b - \lfloor \frac{n}{2b} \rfloor) \cdot \lambda\eta} \quad \text{(by } (1-x)^a \le 1 - ax \text{ for } x > 0 \text{ and } a \ge 1) \quad (145)$$

$$\le \frac{\alpha\eta S_g^2}{4\sigma^2 b^2} \cdot \frac{1}{n/b \cdot \lambda\eta} \quad \text{(by } \frac{n}{b} \ge 2) \quad (146)$$

3. The second term is upper bounded by $\varepsilon_0^1(\alpha) = \frac{\alpha\eta S_g^2}{4\sigma^2 b^2}$, which is smaller than the composition-based privacy bound after $\cdot \frac{n}{b}$ epochs.

## E.2 Proof for Theorem 4.3

*Theorem 4.3.* If the loss function $\ell(\theta; x)$ is $\lambda$-strongly convex, $\beta$-smooth, and if its gradient has finite $\ell_2$-sensitivity $S_g$, then Algorithm 1 with stepsize $\eta < \frac{2}{\lambda+\beta}$ satisfies $(\alpha, \varepsilon)$-Rényi DP guarantee with

$$\varepsilon \leq \frac{1}{\alpha-1} \log\left(S_K^0(\alpha)\right) \tag{147}$$

where the term $S_k^j(\alpha)$ is recursively defined by

$$S_0^0(\alpha) = 1 \tag{148}$$

$$S_k^{j+1}(\alpha) = \frac{b}{n} \cdot e^{\frac{(\alpha-1)\alpha\eta S_g^2}{4\sigma^2 b^2}} \cdot S_k^j(\alpha) + (1 - \frac{b}{n}) \cdot S_k^j(\alpha)^{(1-\eta\lambda)^2}, \text{ for } k = 0, \cdots, K-1 \text{ and } j = 0, \cdots, n/b-1 \tag{149}$$

$$S_{k+1}^0(\alpha) = S_k^{n/b}(\alpha) \text{ for } k = 0, \cdots, K-1 \tag{150}$$

$$\tag{151}$$

*Proof.* By definition,

$$e^{(\alpha-1)\varepsilon_k^{j+1}(\alpha)} = \int \left(\frac{p_k^{j+1}(\theta)}{p'_k^{j+1}(\theta)}\right)^\alpha p'_k^{j+1}(\theta) d\theta \tag{152}$$

$$= \int \left(\frac{\mathbb{E}_{B_0^0, \cdots, B_k^j}\left[p_k^{j+1}(\theta|B_0^0, \cdots, B_k^j)\right]}{\mathbb{E}_{B_0^0, \cdots, B_k^j}\left[p'_k^{j+1}(\theta|B_0^0, \cdots, B_k^j)\right]}\right)^\alpha \cdot \mathbb{E}_{B_0^0, \cdots, B_k^j}\left[p'_k^{j+1}(\theta|B_0^0, \cdots, B_k^j)\right] d\theta \tag{153}$$

By the joint convexity of the function $\frac{x^\alpha}{y^{\alpha-1}}$ on $x, y > 0$ (Lemma 20 Balle et al. [5]), and by the mini-batch decomposition for the distribution of the last iterate parameters in Section 2, we prove

$$e^{(\alpha-1)\varepsilon_k^{j+1}(\alpha)} \leq \mathbb{E}_{B_0^0, \cdots, B_k^j}\left[\int \left(\frac{p_k^{j+1}(\theta|B_0^0, \cdots, B_k^j)}{p'_k^{j+1}(\theta|B_0^0, \cdots, B_k^j)}\right)^\alpha \cdot p'_k^{j+1}(\theta|B_0^0, \cdots, B_k^j) d\theta\right] \tag{154}$$

We now derive a recursive scheme for computing the right hand side term denoted as $S_k^{j+1}(\alpha)$. By definition,

$$S_k^{j+1}(\alpha) = \mathbb{E}_{B_0^0, \cdots, B_k^j}\left[\int \left(\frac{p_k^{j+1}(\theta|B_0^0, \cdots, B_k^j)}{p'_k^{j+1}(\theta|B_0^0, \cdots, B_k^j)}\right)^\alpha \cdot p'_k^{j+1}(\theta|B_0^0, \cdots, B_k^j) d\theta\right] \tag{155}$$

The term inside expectation corresponds to Rényi privacy loss under fixed mini-batches $B_0^0, \cdots, B_k^j$. Therefore, by Lemma 3.2,

$$\int \left(\frac{p_k^{j+1}(\theta|B_0^0, \cdots, B_k^j)}{p'_k^{j+1}(\theta|B_0^0, \cdots, B_k^j)}\right)^\alpha \cdot p'_k^{j+1}(\theta|B_0^0, \cdots, B_k^j) d\theta \tag{156}$$

$$\leq \begin{cases} \int \left(\frac{p_k^j(\theta|B_0^0, \cdots, B_k^j)}{p'_k^j(\theta|B_0^0, \cdots, B_k^j)}\right)^\alpha \cdot p'_k^j(\theta|B_0^0, \cdots, B_k^j) d\theta \cdot e^{\frac{(\alpha-1)\alpha\eta S_g^2}{4\sigma^2 b^2}} & \text{if } i_0 \in B_k^j \\ \left(\int \left(\frac{p_k^j(\theta|B_0^0, \cdots, B_k^j)}{p'_k^j(\theta|B_0^0, \cdots, B_k^j)}\right)^\alpha \cdot p'_k^j(\theta|B_0^0, \cdots, B_k^j) d\theta\right)^{(1-\eta\lambda)^2} & \text{if } i_0 \notin B_k^j \end{cases} \tag{157}$$

By plugging Equation (157), and by the definition of $S_k^{j+1}(\alpha)$, we prove

$$S_k^{j+1}(\alpha) = \mathbb{E}_{B_0^0, \cdots, B_k^j} \left[ \int \left( \frac{p_k^{j+1}(\theta | B_0^0, \cdots, B_k^j)}{p'^{j+1}_k(\theta | B_0^0, \cdots, B_k^j)} \right)^\alpha \cdot p'^{j+1}_k(\theta | B_0^0, \cdots, B_k^j) d\theta \right] \tag{158}$$

$$= P(i_0 \in B_k^j) \cdot S_k^j \cdot e^{\frac{(\alpha-1)\alpha \eta S_g^2}{4\sigma^2 b^2}} + P(i_0 \notin B_k) \cdot \tag{159}$$

$$\mathbb{E}_{B_0^0, \cdots, B_k^{j-1}} \left[ \left( \int \left( \frac{p_k^j(\theta | B_0^0, \cdots, B_k^j)}{p'^j_k(\theta | B_0^0, \cdots, B_k^j)} \right)^\alpha \cdot p'^j_k(\theta | B_0^0, \cdots, B_k^j) d\theta \right)^{(1-\eta\lambda)^2} \right] \tag{160}$$

By further using the concavity of $x^{(1-\eta\lambda)^2}$, and the definition of $S_k^j(\alpha)$, we prove

$$S_k^{j+1}(\alpha) \leq P(i_0 \in B_k^j) \cdot S_k^j \cdot e^{\frac{(\alpha-1)\alpha \eta S_g^2}{4\sigma^2 b^2}} + P(i_0 \notin B_k) \cdot \tag{161}$$

$$\left( \mathbb{E}_{B_0^0, \cdots, B_k^{j-1}} \left[ \int \left( \frac{p_k^j(\theta | B_0^0, \cdots, B_k^j)}{p'^j_k(\theta | B_0^0, \cdots, B_k^j)} \right)^\alpha \cdot p'^j_k(\theta | B_0^0, \cdots, B_k^j) d\theta \right] \right)^{(1-\eta\lambda)^2} \tag{162}$$

$$= \frac{b}{n} \cdot e^{\frac{(\alpha-1)\alpha \eta S_g^2}{4\sigma^2 b^2}} \cdot S_k^j + \left( 1 - \frac{b}{n} \right) \cdot (S_k^j)^{(1-\eta\lambda)^2} \tag{163}$$

Therefore, the full recursive scheme for computing $S_k^j(\alpha)$ is as follows.

1. $S_0^0(\alpha) = 1$

2. $S_k^{j+1}(\alpha) = \frac{b}{n} \cdot e^{\frac{(\alpha-1)\alpha \eta S_g^2}{4\sigma^2 b^2}} \cdot S_k^j(\alpha) + (1 - \frac{b}{n}) \cdot S_k^j(\alpha)^{(1-\eta\lambda)^2}$, for $k = 0, \cdots, K - 1$ and $j = 0, \cdots, n/b - 1$.

3. $S_{k+1}^0(\alpha) = S_k^{n/b}(\alpha)$ for $k = 0, \cdots, K - 1$.

And Algorithm 1 satisfies $(\alpha, \varepsilon)$-Rényi DP guarantee with

$$\varepsilon \leq \frac{1}{\alpha - 1} \log \left( S_K^0(\alpha) \right) \tag{164}$$

$\square$

# F    Proofs and Explanations for Section 5

## F.1    Pseudocode for DP-SGD under notations in this paper

In Algorithm 2, we provide the pseudocode of an equivalent of DP-SGD algorithm under notations in our paper. For the convenience of implementation in existing privacy libaries, we introduce the noise multiplier $\sigma_{mul}$ to substitute $\sigma$ in Algorithm 1. We comment that the noisy gradient update in Algorithm 2 under noise multiplier $\sigma_{mul}$, is equivalent to a noisy gradient update in Algorithm 1 with noise standard deviation $\sigma = \sqrt{\frac{\eta}{2}} \cdot \frac{1}{b} \cdot \sigma_{mul} \cdot \frac{S_g}{2}$. By plugging this equivalent $\sigma$ into Theorem 4.2, we prove the privacy dynamics bound in Corollary 5.3 for regularized logistic regression.

## F.2    Proof for ensuring strong convexity

***Regularized Logistic regression (for strong convexity).*** The loss function for regularized logistic regression in the multi-class setting (with per-class bias) is as follows.

$$\ell_\lambda(\theta; \mathbf{x}, \mathbf{y}) = \ell_0(\theta; \mathbf{x}, \mathbf{y}) + \frac{\lambda}{2} \|\theta\|_2^2 \tag{165}$$

---
**Algorithm 2** $\mathcal{A}_{\text{implementation}}$: Noisy mini-batch Gradient Descent on regularized logistic regression loss function

---
**Input:** Data domain $\mathcal{X}$. Dataset $D = ((\mathbf{x}_1, \mathbf{y}_1), (\mathbf{x}_2, \mathbf{y}_2), \cdots, (\mathbf{x}_n, \mathbf{y}_n))$, where each data point consists of the feature vector $\mathbf{x}_i \in \mathbb{R}^d$ and the label vector $\mathbf{y}_i \in \{0, 1\}^c$. The logistic regression loss function $\ell_0(\theta; \mathbf{x}, \mathbf{y})$ defined as Equation (8) with parameter space $\mathbb{R}^{(d+1) \cdot c}$. Stepsize $\eta$, noise multiplier $\sigma_{mul}$, a (data-independent) parameter initialization distribution $p_0(\theta)$, mini-batch size $b$, feature clipping norm $L$, and (unregularized) gradient clipping norm $\frac{S_g}{2}$.

**Feature Normalization:** $\mathbf{x}_1, \cdots, \mathbf{x}_n \leftarrow \text{normalize}(\mathbf{x}_1, \cdots, \mathbf{x}_n)$, where normalize() is an $(\alpha, \varepsilon_{norm})$-Rényi differentially private batch normalization or group normalization scheme described in Tramèr and Boneh [34] (Section 2.3 and Appendix B).

**Feature Clipping:** $\mathbf{x}_i \leftarrow \frac{\mathbf{x}_i}{\|\mathbf{x}_i\|_2} \cdot \min\{\|\mathbf{x}_i\|_2, L\}$.

**Initialization:** Sample $\theta_0^0$ from the initialization distribution $p_0(\theta)$.

**Batch Generation:** shuffle the indices set $\{1, \cdots, n\}$, and partition them into $n/b$ sequential mini-batches $B^0, \cdots, B^{n/b-1}$ that are subsets of $\{1, \cdots, n\}$, each with size $b$.

**for** $k = 0, 1, \cdots, K - 1$ **do**

    **for** $j = 0, 1, \cdots, n/b - 1$ **do**

        **Gradient Clipping:** $g_0(\theta_k^j, B^j) = \frac{1}{b} \sum_{x_i \in B^j} \frac{\nabla \ell(\theta_k^j; \mathbf{x}_i)}{\|\nabla \ell_0(\theta_k^j; \mathbf{x}_i)\|_2} \cdot \min\{\|\nabla \ell_0(\theta_k^j; \mathbf{x}_i)\|_2, \frac{S_g}{2}\}$

        **Regularization:** $g\left(\theta_k^j; B^j\right) = g_0(\theta_k^j; B^j) + \lambda \cdot \theta_k^j$

        $\theta_k^{j+1} = \theta_k^j - \eta \cdot g\left(\theta_k^j; B^j\right) + \eta \cdot \frac{1}{b} \cdot \sigma_{mul} \cdot \frac{S_g}{2} \cdot \mathcal{N}\left(0, \mathbb{I}_d\right)$

    $\theta_{k+1}^0 = \theta_k^{n/b}$

Output $\theta_K^0$

---

where $\ell_0(\theta; \mathbf{x}, \mathbf{y})$ is the following logistic regression loss function.

$$\ell_0(\theta; \mathbf{x}, \mathbf{y}) = -\mathbf{y}^1 \log\left(\frac{e^{\bar{x}^T \cdot \theta_1}}{e^{\bar{x}^T \cdot \theta_1} + \cdots + e^{\bar{x}^T \cdot \theta_c}}\right) - \cdots - \mathbf{y}^c \log\left(\frac{e^{\bar{x}^T \cdot \theta_c}}{e^{\bar{x}^T \cdot \theta_1} + \cdots + e^{\bar{x}^T \cdot \theta_c}}\right) \quad (166)$$

where $\bar{\mathbf{x}} = (\mathbf{x}, 1) \in \mathbb{R}^{d+1}$ denotes the concatenation of the data feature vector $\mathbf{x}$ and 1, and $\mathbf{y} = (\mathbf{y}^1, \cdots, \mathbf{y}^c)$ is the label vector. The parameter vector is $\theta = (\theta_1, \cdots, \theta_c) \in \mathbb{R}^{(d+1) \cdot c}$ that represents the weight and the per-class bias of the linear model. The logistic regression loss function is convex, and therefore the regularized logistic regression loss function is $\lambda$-strongly convex.

### F.3 Proof for ensuring smoothness

***Feature Normalization (for faster convergence).*** For a better convergence of the learning task, we follow Tramèr and Boneh [34], and use feature normalization (including both batch normalization and group normalization). Batch normalization first computes the per-channel mean and variance of the training dataset, in a differentially private way, and then normalizes each channel of each data point in the training dataset. Group normalization [38] separates the channels in a data feature vector into a number of different groups, and normalizes each channel of each data point with the per-point per-group mean and variance (which does not incur additional privacy cost). We refer the reader to [34] (their Section 2.2 and Appendix B) for more details.

***Feature Clipping (for bounding the smoothness constant).*** To ensure that the condition of finite gradient sensitivity $S_g$ is satisfied in our experiments, we follow Feldman et al. [18] and normalize the data feature vector in $\ell_2$ norm, such that $\|\mathbf{x}\|_2 \leq L$. Under this data feature clipping, we prove that the logistic regression loss function (8) is $(\frac{L^2+1}{2})$-smooth in the following Proposition 5.1.

**Proposition F.1.** *If the data feature vector $\mathbf{x}$ has bounded $\ell_2$ norm, such that $\|\mathbf{x}\|_2 \leq L$, then the unregularized logistic regression loss function $\ell_0(\theta; \mathbf{x}, \mathbf{y})$ Equation (8) is convex , $L$-Lipschitz and $\beta$-smooth with regard to parameters $\theta$, for*

$$L = \sqrt{2(L^2 + 1)} \quad (167)$$

$$\beta = \frac{L^2 + 1}{2} \quad (168)$$

*Proof.* We compute the gradient and Hessian matrix for the logistic regression loss function as follows.

The gradient of the logistic regression loss function Equation (8) with respect to $\theta$ is:

$$\nabla_\theta \ell_0(\theta; \mathbf{x}, \mathbf{y}) = -\mathbf{y}^1 \cdot \nabla_\theta \log \left( \frac{e^{\bar{\mathbf{x}}^T \cdot \theta_1}}{e^{\bar{\mathbf{x}}^T \cdot \theta_1} + \cdots + e^{\bar{\mathbf{x}}^T \cdot \theta_c}} \right) - \cdots - \mathbf{y}^c \cdot \nabla_\theta \log \left( \frac{e^{\bar{\mathbf{x}}^T \cdot \theta_c}}{e^{\bar{\mathbf{x}}^T \cdot \theta_c} + \cdots + e^{\bar{\mathbf{x}}^T \cdot \theta_c}} \right) \tag{169}$$

$$= \left( \left( \frac{e^{\bar{\mathbf{x}}^T \cdot \theta_1}}{e^{\bar{\mathbf{x}}^T \cdot \theta_1} + \cdots + e^{\bar{\mathbf{x}}^T \cdot \theta_c}} - \mathbf{y}^1 \right) \bar{\mathbf{x}}^T, \cdots, \left( \frac{e^{\bar{\mathbf{x}}^T \cdot \theta_1}}{e^{\bar{\mathbf{x}}^T \cdot \theta_c} + \cdots + e^{\bar{\mathbf{x}}^T \cdot \theta_c}} - \mathbf{y}^c \right) \bar{\mathbf{x}}^T \right) \tag{170}$$

$$\tag{171}$$

The Hessian of loss function with respect to $\theta$ is:

$$\nabla_\theta^2 \ell_0(\theta; \mathbf{x}, \mathbf{y}) = \begin{pmatrix} H_{11} & \cdots & H_{1c} \\ \vdots & \vdots & \vdots \\ H_{c1} & \cdots & H_{cc} \end{pmatrix} \tag{172}$$

where the submatrices are as follows.

$$\text{For } i = j, \quad H_{ii} = \frac{e^{\bar{\mathbf{x}}^T \cdot \theta_i}}{e^{\bar{\mathbf{x}}^T \cdot \theta_1} + \cdots + e^{\bar{\mathbf{x}}^T \cdot \theta_c}} \cdot \left( 1 - \frac{e^{\bar{\mathbf{x}}^T \cdot \theta_i}}{e^{\bar{\mathbf{x}}^T \cdot \theta_1} + \cdots + e^{\bar{\mathbf{x}}^T \cdot \theta_c}} \right) \bar{\mathbf{x}} \cdot \bar{\mathbf{x}}^T \tag{173}$$

$$\text{For } i \neq j, \quad H_{ij} = -\frac{e^{\bar{\mathbf{x}}^T \cdot \theta_i}}{e^{\bar{\mathbf{x}}^T \cdot \theta_1} + \cdots + e^{\bar{\mathbf{x}}^T \cdot \theta_c}} \cdot \frac{e^{\bar{\mathbf{x}}^T \cdot \theta_j}}{e^{\bar{\mathbf{x}}^T \cdot \theta_1} + \cdots + e^{\bar{\mathbf{x}}^T \cdot \theta_c}} \cdot \bar{\mathbf{x}} \cdot \bar{\mathbf{x}}^T \tag{174}$$

$$\tag{175}$$

Therefore, the Hessian matrix of the loss function with respect to $\theta$ equals the following Kronecker product.

$$\nabla_\theta^2 \ell_0(\theta; \mathbf{x}, \mathbf{y}) = T \otimes (\bar{\mathbf{x}} \cdot \bar{\mathbf{x}}^T) \tag{176}$$

$$\text{where } T_{ij} = \begin{cases} \frac{e^{\bar{\mathbf{x}}^T \cdot \theta_i}}{e^{\bar{\mathbf{x}}^T \cdot \theta_1} + \cdots + e^{\bar{\mathbf{x}}^T \cdot \theta_c}} \cdot \left( 1 - \frac{e^{\bar{\mathbf{x}}^T \cdot \theta_i}}{e^{\bar{\mathbf{x}}^T \cdot \theta_1} + \cdots + e^{\bar{\mathbf{x}}^T \cdot \theta_c}} \right) & i = j \\ -\frac{e^{\bar{\mathbf{x}}^T \cdot \theta_i}}{e^{\bar{\mathbf{x}}^T \cdot \theta_1} + \cdots + e^{\bar{\mathbf{x}}^T \cdot \theta_c}} \cdot \frac{e^{\bar{\mathbf{x}}^T \cdot \theta_j}}{e^{\bar{\mathbf{x}}^T \cdot \theta_1} + \cdots + e^{\bar{\mathbf{x}}^T \cdot \theta_c}} & i \neq j \end{cases} \tag{177}$$

By Böhning [9], $T$ is a positively semi-definite matrix that satisfies

$$T \preceq \frac{1}{2} \cdot \left( I_c - \frac{1}{c} \cdot \mathbf{1}_c \cdot \mathbf{1}_c^T \right), \text{ where } \mathbf{1}_c = (1, \cdots, 1)^T \in \mathbb{R}^c. \tag{178}$$

Therefore, the eigenvalues of $T$ fall in the range $[0, \frac{1}{2}]$. Therefore, because the eigenvalues for the Kronecker product matrix $\nabla_\theta^2 \ell_0(\theta; \mathbf{x}, \mathbf{y}) = T \otimes (\bar{\mathbf{x}} \cdot \bar{\mathbf{x}}^T)$ consists of the product of eigenvalues for $T$ and $\bar{\mathbf{x}} \cdot \bar{\mathbf{x}}^T$, we prove that the eigenvalues for $\nabla_\theta^2 \ell_0(\theta; \mathbf{x}, \mathbf{y})$ fall in the range of $[0, \frac{1}{2}\|\bar{\mathbf{x}}\|_2^2]$.

By $\|\mathbf{x}\|_2 \leq L$, we prove that $\|\bar{\mathbf{x}}\|_2 \leq \sqrt{L^2 + 1}$, therefore Equation (9) and Equation (10) (in the proposition statement) hold. $\square$

These computations of these Lipschitz smoothness constants could also be cross-checked in the tensorflow privacy tutorial for logistic regression. https://github.com/tensorflow/privacy/blob/master/tutorials/mnist_lr_tutorial.py. The feature clipping technique is different from the DP-SGD [1] algorithm that only requires per-example gradient clipping. The major reason that we use data feature clipping (besides per-example gradient clipping), is for ensuring smoothness of the logistic regression loss function (by Proposition 5.1), which is a necessary condition for applying our privacy bound Theorem 3.3.

### F.4 Proof for ensuring finite gradient sensitivity

***Per-example Clipping on Unregularized Gradient (for reducing gradient sensitivity without harming smoothness or strong convexity).*** Although feature clipping already bounds the gradient sensitivity by $2\sqrt{2(L^2+1)}$ (by Proposition 5.1), this bound grows with the feature clipping norm $L$. This in turn restricts the signal to noise ratio, and does not give good empirical privacy-utility trade-off in our experiments. Therefore, we additionally perform per-example $\ell_2$-clipping on the unregularized gradient (detailed pseudocode in Appendix F.1). Under per-example clipping on unregularized gradient, we prove in the following Proposition 5.2, that each gradient update in *regularized logistic regression* has finite gradient sensitivity, and preserves strong convexity and smoothness.

**Proposition F.2.** *Let $\ell_0(\theta; \mathbf{x}, \mathbf{y})$ be the logistic regression loss function defined in Equation (8). Let $g_0(\theta; \mathbf{x}, \mathbf{y}) = \frac{\nabla \ell_0(\theta; \mathbf{x}, \mathbf{y})}{\|\nabla \ell_0(\theta; \mathbf{x}, \mathbf{y})\|_2} \cdot \min\{\|\nabla \ell_0(\theta; \mathbf{x}, \mathbf{y})\|_2, \frac{S_g}{2}\}$ be the clipped gradient of (unregularized) loss function $\ell_0(\theta; \mathbf{x}, \mathbf{y})$, under $\ell_2$ clipping norm $\frac{S_g}{2}$. If $g(\theta; \mathbf{x}, \mathbf{y}) = g_0(\theta; \mathbf{x}, \mathbf{y}) + \lambda \theta$, and if the data vector $\mathbf{x}$ has bounded $\ell_2$ norm, such that $\|\mathbf{x}\|_2 \leq L$, then $g(\theta; \mathbf{x}, y)$ has finite $\ell_2$-sensitivity $S_g$, is continuous, and is almost everywhere differentiable with*

$$\lambda \cdot \mathbb{I}_{(d+1)\cdot c} \preceq \nabla_\theta g(\theta; \mathbf{x}, \mathbf{y}) \preceq (\beta + \lambda) \cdot \mathbb{I}_{(d+1)\cdot c} \tag{179}$$

*for any $\theta, \theta' \in \mathbb{R}^{(d+1)\cdot c}$ and $\beta = \frac{L^2+1}{2}$.*

We provide complete proof for this proposition below. This construction of clipped unregularized gradient facilitates us to enjoy the benefits of gradient clipping (such as for speeding up convergence [41, 14]) while satsifying the necessary smoothness and strong convexity conditions for applying our privacy dynamics bound.

*Proof.* By definition Equation (8) for the logistic regression loss function, $\nabla \ell_0(\theta; \mathbf{x}, \mathbf{y})$ is twice continuously differentiable and convex. By Proposition 5.1, $\ell_\lambda(\theta; \mathbf{x}, \mathbf{y})$ is $\beta$-smooth with $\beta = \frac{L^2+1}{2}$. Therefore, the Hessian matrix of $\ell_0(\theta; \mathbf{x}, \mathbf{y})$ satisfies the following inequality.

$$0 \cdot \mathbb{I}_{(d+1)\cdot c} \preceq \nabla_\theta^2 \ell_0(\theta; \mathbf{x}, \mathbf{y}) \preceq \beta \cdot \mathbb{I}_{(d+1)\cdot c}, \tag{180}$$

where $d$ is the dimension of the input data feature vector $\mathbf{x}$, and $c$ is the number of classes in the label vector $\mathbf{y}$. Moreover, the clipped (unregularized) gradient $g_0(\theta; \mathbf{x}, \mathbf{y})$ (under $\ell_2$ clipping norm $\frac{S_g}{2}$) is continuous, and is almost everywhere differentiable as follows.

$$\nabla_\theta g_0(\theta; \mathbf{x}, \mathbf{y}) =$$
$$\begin{cases} \nabla_\theta^2 \ell_0(\theta; \mathbf{x}, \mathbf{y}) & \text{if } \|\nabla_\theta \ell_0(\theta; \mathbf{x}, \mathbf{y})\|_2 < \frac{S_g}{2} \\ \frac{S_g}{2} \cdot \frac{1}{\|\nabla_\theta \ell_0(\theta; \mathbf{x}, \mathbf{y})\|_2} \cdot M \cdot \nabla_\theta^2 \ell_0(\theta; \mathbf{x}, \mathbf{y}) & \text{if } \|\nabla_\theta \ell_0(\theta; \mathbf{x}, \mathbf{y})\|_2 > \frac{S_g}{2} \end{cases} \tag{181}$$

where $M = \mathbb{I}_{(d+1)\cdot c} - \frac{\nabla \ell_0(\theta; \mathbf{x}, \mathbf{y})}{\|\nabla \ell_0(\theta; \mathbf{x}, \mathbf{y})\|_2} \cdot \left(\frac{\nabla \ell_0(\theta; \mathbf{x}, \mathbf{y})}{\|\nabla \ell_0(\theta; \mathbf{x}, \mathbf{y})\|_2}\right)^T$ is a symmetric matrix. Because $\frac{\nabla \ell_0(\theta; \mathbf{x}, \mathbf{y})}{\|\nabla \ell_0(\theta; \mathbf{x}, \mathbf{y})\|_2}$ is a unit vector, we prove that $M$ is positive semi-definite and satisfies $0 \cdot \mathbb{I}_{(d+1)\cdot c} \preceq M \preceq \mathbb{I}_{(d+1)\cdot c}$. Moreover, for the case where $\|\nabla_\theta \ell_0(\theta; \mathbf{x}, \mathbf{y})\|_2 > \frac{S_g}{2}$, we have that $\frac{S_g}{2} \cdot \frac{1}{\|\nabla_\theta \ell_0(\theta; \mathbf{x}, \mathbf{y})\|_2} \leq 1$. By combining this ineuality with property of $M$ and Equation (180), we prove that for any $\theta$ such that $\|\nabla_\theta \ell(\theta; \mathbf{x}, \mathbf{y})\|_2 \neq \frac{S_g}{2}$,

$$0 \cdot \mathbb{I}_{(d+1)\cdot c} \preceq \nabla_\theta g_0(\theta; \mathbf{x}, \mathbf{y}) \preceq \beta \cdot \mathbb{I}_{(d+1)\cdot c} \tag{182}$$

Therefore, by plugging this into the definition of $g(\theta; \mathbf{x}, \mathbf{y}) = g_0(\theta; \mathbf{x}, \mathbf{y}) + \lambda \theta$, we prove that $g(\theta; \mathbf{x}, \mathbf{y})$ is continous, and is almost everywhere differentiable with

$$\lambda \cdot \mathbb{I}_{(d+1)\cdot c} \preceq \nabla_\theta g(\theta; \mathbf{x}, \mathbf{y}) \preceq (\beta + \lambda) \cdot \mathbb{I}_{(d+1)\cdot c} \tag{183}$$

$\square$