# OpenReview forum: "Differentially Private Learning Needs Hidden State (Or Much Faster Convergence)"
_NeurIPS.cc/2022/Conference — NeurIPS 2022 Accept_

### Official Review · Reviewer_Wz9m · 2022-07-11

**Rating:** 7
**Confidence:** 3
**Soundness:** 4 excellent
**Presentation:** 3 good
**Contribution:** 3 good

**Summary:**

In this paper, authors extend recent work in convergent privacy cost of convex optimization to subsampled gradient descent (GD) setting. Authors show, that for strongly convex Lipschitz loss functions, the privacy cost of releasing last iterate of DP-SGD does not grow to infinity as the number of epochs grows, but that there is a finite upper bound for this loss. Authors present a novel privacy analysis that builds on top of recent work by Chourasia et al. 2021, and is able to capture the privacy amplification by subsampling. Finally, authors demonstrate empirically how the reduced privacy cost improves the prediction accuracy of logistic regression model learned under DP.

**Questions:**

Typos:
- Lines 108-109: sentence inside parenthesis seems bit odd. Also, missing capital letter after period as in ". then"
- Thm 5.2: I guess K>=1 being mentioned twice is a typo.

Some questions:

- When you say "fixed sequence of minibatches", does this mean that the minibatches are the same across the epochs? As in $B^j_k = B^j_l \, \forall k,l \in [K]$? I just want to make sure that I understand the difference in the contributions of Sections 4 and 5. So Sec. 4 builds the theoretical background under the assumption that the sequence is fixed and Sec. 5 then extends this to actual subsampling scheme?

- In beginning of Sec. 4, it sounds like you say that you focus on the case where the adjacent index is not accessed by Alg. 1. This seems weird given the theoretical contributions of this Section, e.g. Thms 4.2 and 4.3. Maybe it is a typo? Or did I just misunderstand your point.

- Out of curiosity, could you use some learning rate adaptation like Adam under the last iterate release scheme? I wonder would it cause some side-channel privacy leak through the gradient history it uses to update the learning rate.

- Maybe I missed something, but after Thm 5.2 you mention that the bound is tighter than composition based after $4 \cdot \frac{1}{\lambda  \eta} + 4 \cdot \frac{n}{b}$ epochs. You justify this by saying, that the parts leading to this bound are smaller than the composition based one after $4 \cdot \frac{1}{\lambda  \eta}$ epochs and $4 \cdot \frac{n}{b}$ epochs. Now, I wonder, why cannot you just pick the max of these and say that the both parts are majored with the composition based after these many epochs. Of course the sum of the terms also works, but I was just curious if there is a reason not to use $\max(4 \cdot \frac{1}{\lambda  \eta}, 4 \cdot \frac{n}{b}) \leq 4 \cdot \frac{1}{\lambda  \eta} + 4 \cdot \frac{n}{b}$?

**Limitations:**

I think the limited use cases due to assumptions on the loss function could be further discussed. Negative societal impact is not discussed in the paper, but I believe the paper is purely theoretical and does not have direct negative impacts.

**Strengths And Weaknesses:**

In many applications, the intermediate steps of DP-SGD are not needed, but are released which causes the privacy cost of these types of release schemes to grow as the number of iterations grows. In this paper, authors continue the research in direction of releasing only the last iterate and quantifying the privacy amplification due to keeping the internal states secret. More specifically, this paper extends the work by Chourasia et al. 2021, who proposed a bound for the privacy cost for DP gradient descent algorithm. As the GD algorithm is often computationally expensive, authors take the natural next step and quantify the privacy cost of stochastic GD. Authors show several novel theoretical results (mainly Thms 4.2, 4.3, 5.2 and 5.3), that are based on novel privacy analysis, capturing the privacy amplifying effect of subsampling. The new privacy bound is significantly tighter than the privacy bound of Chourasia et al. that does not capture the subsampling amplification.

I also think that the paper is generally well written, and after couple of reads the story and the contributions become very clear. There are some parts that I think could be further clarified, and I will add those to the questions section.

The most significant limitation of this work must be the assumptions on the loss function. The strong convexity and the Lipschitzity rule out quite a few applications of DP-SGD. However, I still believe that the paper makes significant contributions that will be beneficial in the future works on convergent bounds of DP-SGD.

---

> ### Author Response · Authors · 2022-08-02
> **Response to Reviewer Wz9m (2/2)**
>
>
> **[Question 4]**
>
> > ... after Thm 5.2 you mention that the bound is tighter than composition based after $4\cdot \frac{1}{\lambda\eta} + 4 \cdot \frac{n}{b}$ epochs ... Of course the sum of the terms also works, but I was just curious if there is a reason not to use $\max(4\cdot \frac{1}{\lambda\eta}, 4\cdot \frac{n}{b})\leq 4\cdot \frac{1}{\lambda\eta} + 4\cdot \frac{n}{b}$?
>
> We did not use $\max(4\cdot \frac{1}{\lambda\eta}, 4\cdot \frac{n}{b})$ because it does not necessarily ensure our bound in Theorem 5.2 (which is the sum of two terms)  to be smaller than composition-based bound. Instead, it only ensures that each of the two terms alone is smaller than the composition-based bound. Consequently, after $\max(4\cdot \frac{1}{\lambda\eta}, 4\cdot \frac{n}{b})$ epochs, we could only prove that our bound is smaller than **two** times the composition-based privacy bound.
>
> On the contrary, after $4\cdot \frac{1}{\lambda\eta} + 4\cdot \frac{n}{b}$ epochs, we could prove that our bound is smaller than composition-based bound as follows.
> - For convenience, we denote our converged privacy bound in Theorem 5.2 as A + B, where A is the first term, and B is the second term. We denote the composition-based R\'enyi DP bound after $K$ epochs as $C_K$. Then in Appendix D.1, we have proved that $A\leq C_{4\cdot \frac{1}{\lambda\eta}}$ and $B\leq C_{4\cdot \frac{n}{b}}$. Therefore, our converged privacy bound $A + B\leq C_{4\cdot \frac{1}{\lambda\eta}} + C_{4\cdot \frac{n}{b}}$.
> - Because composition-based R\'enyi DP bound $C_K$ is linearly growing with the number of epochs $K$, we prove that $C_{4\cdot \frac{1}{\lambda\eta}} + C_{4\cdot \frac{n}{b}} = C_{4\cdot \frac{1}{\lambda\eta} + 4\cdot \frac{n}{b}}$.
> - Finally, by combining the above two steps, we prove that our converged privacy bound $A+B\leq C_{4\cdot \frac{1}{\lambda\eta} + 4\cdot \frac{n}{b}}$.
>
> With slight modification, however, there is indeed an alternative bound that takes maximum over two terms (similar to what you are proposing): our bound is smaller than composition-based bound after $\max(8\cdot \frac{1}{\lambda\eta}, 8\cdot \frac{n}{b})$ epochs.
> - By using Appendix 5.1 and the linearity of composition-based R\'enyi DP bound with regard to number of epochs $K$, we prove that the first term $A\leq \frac{1}{2}\cdot C_{2\cdot 4\cdot \frac{1}{\lambda\eta}}$ and the second term $B\leq \frac{1}{2}\cdot C_{2\cdot 4\cdot \frac{n}{b}}$.
> - This implies that $A,B\leq \frac{1}{2}C_{K}$ after $K=\max(8\cdot \frac{1}{\lambda\eta}, 8\cdot \frac{n}{b})$ epochs, because the composition-based privacy bound $C_{K}$ is monotonically increasing with the number of epochs $K$.
> - Therefore, by combining the two inequalities $A\leq \frac{1}{2}C_K$ and $B\leq \frac{1}{2}C_K$, we prove that $A+B\leq C_K$, for $K=\max(8\cdot \frac{1}{\lambda\eta}, 8\cdot \frac{n}{b})$.
>
> However, this alternative bound $\max(8\cdot \frac{1}{\lambda\eta}, 8\cdot \frac{n}{b})$ is strictly worse than the bound that we proved $4\cdot \frac{1}{\lambda\eta} + 4\cdot \frac{n}{b}$. Therefore, in this paper, we stick to $4\cdot \frac{1}{\lambda\eta} + 4\cdot \frac{n}{b}$ as a bound for the number of epochs.

---

> ### Author Response · Authors · 2022-08-02
> **Response to Reviewer Wz9m (1/2)**
>
>
> > Typos in line 108-109
>
> Thanks for pointing out these typos. We have corrected lines 108-109 in the revised paper.
>
> **[Question 1]**
>
> > When you say "fixed sequence of minibatches", does this mean that the minibatches are the same across the epochs? As in $B_k^j=B_l^j, \forall l\in [K]$? I just want to make sure that I understand the difference in the contributions of Sections 4 and 5. So Sec. 4 builds the theoretical background under the assumption that the sequence is fixed and Sec. 5 then extends this to actual subsampling scheme?
>
> Yes, it is correct.
>
> **[Question 2]**
>
> > In beginning of Sec. 4, it sounds like you say that you focus on the case where the adjacent index is not accessed by Alg. 1. This seems weird given the theoretical contributions of this Section, e.g. Thms 4.2 and 4.3. Maybe it is a typo? Or did I just misunderstand your point.
>
> Thank you for the comment. We've updated the beginning of Section 4 to further clarify our points.
>
> The main contribution of Section 4 is the privacy amplification when the adjacent index **is not** accessed by Alg. 1 (i.e., randomized post-processing). This amplification is quantified in the first row of Lemma 4.2, and a small amplification ratio is the key to proving a converging privacy bound.
>
> On the other hand, we do also quantify the privacy loss growth when the adjacent index **is** accessed by Alg. 1. However, this is not the main contribution of Section 4. This is because, as shown in the second row of Lemma 4.2, we are simply using the standard R\'enyi DP composition tool to quantify this privacy loss growth.
>
> **[Question 3]**
>
> > Out of curiosity, could you use some learning rate adaptation like Adam under the last iterate release scheme? I wonder would it cause some side-channel privacy leak through the gradient history it uses to update the learning rate.
>
> Thank you for raising this interesting open problem. We do not have a full answer yet, but we are happy to share our thoughts.
>
> On the one hand, our hidden-state privacy analysis for noisy SGD easily extends to stepsize scaling (such as logarithmically or polynomially decaying stepsize). On the other hand, we believe that extending hidden-state privacy bound to adaptive update schemes that use past gradient information (such as SGD with momentum, Adagrad and Adam) requires non-trivial new efforts. This is due to the following reasons.
> - Each state in Adam algorithm not only contains the model parameters $\theta_K$, but also contains a summary of past gradient information up to iteration $K$ (e.g., the first order moment $\mu_K$ and second order moment $m_K$).
> - Consequently, the Lipschitz constant $L$ for each Adam update $(\theta_k, \mu_k, m_k)\rightarrow (\theta_{k+1}, \mu_{k+1}, m_{k+1})$, as well as the log-Sobolev inequality constant for the distribution of intermediate state $(\theta_{k+1}, \mu_{k+1}, m_{k+1})$, would become significantly more difficult to analyze, when compared to the analysis under vanilla noisy (S)GD updates. This is also the main difficulty for applying our Lemma 4.2 to derive hidden-state privacy bound for Adam.
> - However, once the Lipschitz constant $L$ for each Adam update and the log-Sobolev inequality constant for each intermediate state $(\theta_k,\mu_k,m_k)$ are proved, we believe that our Lemma 4.2 is applicable for deriving recursive hidden-state privacy bound. (Whether this hidden-state privacy bound converges, however, further depends on how good the derived log-Sobolev inequality constants are for the intermediate states of the algorithm. We refer to our response to Reviewer duNW [Question 1] for more discussion.)

---

### Official Review · Reviewer_tVnr · 2022-07-11

**Rating:** 4
**Confidence:** 2
**Soundness:** 2 fair
**Presentation:** 2 fair
**Contribution:** 2 fair

**Summary:**

This work improves the analysis of the last-iterate privacy bound which assumes the internal state of the algorithm is not revealed. The new analysis derives a privacy loss that grows slower than the one derived from the composition theorem if the network converges slowly or is trained for multiple epochs.

**Questions:**

Q1: Although the idea of sparing privacy by hiding internal states looks interesting, many assumptions have been made for the analysis, which could be broken in practice and thus breach privacy, on the other hand, the evaluation does not show the proposed method obviously outperforms the composition theorem (as described in Weakness), thus one may lack the motivation for using the proposed method and taking a risk.

Q2: I don't know how the hyperparameters are tuned, but since the privacy loss of the last-iterate analysis becomes flat after multiple epochs of training, perhaps longer training will make the results of the proposed method better.

**Strengths And Weaknesses:**

I have a basic understanding of the theoretical part of this work but didn't go deeper into the proofs, for me the strength and weakness of this work are:

Strength:

So far DP learning still suffers from dramatic performance loss compared with non-private learning, tailored and tight privacy bound can help reduce the required noise under certain privacy budgets and improve the performance.

Weaknesses:

1. This work extends the study of last-iterate privacy bound by considering mini-batch gradient and privacy amplification by subsampling, etc.. To apply the new analysis in practice, many assumptions have been made, which may not be satisfied when training a real network, so I expect the resulting network obviously outperforms others, so far it's not, perhaps a tighter bound is needed.

2. I am a little confused about the evaluation of the proposed privacy analysis. I see the baseline performance given in this work is lower than in its original work. For example in the experiment on CIFAR10, ScatterNet can actually obtain an accuracy of 69.3\%, ResNeXt-29 can achieve 80.0\%, SIMCLR can achieve 92.7\%, all are on par with using the method proposed in this work.

---

> ### Author Response · Authors · 2022-08-02
> **Response to Reviewer tVnr (2/2)**
>
> **[Question 2]**
> > I don't know how the hyperparameters are tuned, but since the privacy loss of the last-iterate analysis becomes flat after multiple epochs of training, perhaps longer training will make the results of the proposed method better.
>
> **Regarding hyper-parameter tuning details:** we report the hyperparameters tuning process in Appendix E.6-E.8. Please also see our attached code `run_baselines_cifar10_feature_clip.py` for an example hyperparameter tuning script for cifar10.
>
> **Regarding the suggestion of using longer training process to make the results better:** Yes, you are right. And longer training is exactly what we are doing in our experiments to enable better accuracy under our hidden-state privacy analysis (than accuracy under composition-based privacy analysis). More specifically, the optimal accuracy under privacy dynamics analysis is reached at above 1200 epochs for training linear models on handcrafted features (600 epochs for fine-tuning). Meanwhile, DP-SGD achieves the optimal accuracy at below 60 epochs for training on handcrafted features (40 epochs for fine-tuning) under composition-based privacy bound.

---

> ### Author Response · Authors · 2022-08-02
> **Response to Reviewer tVnr (1/2)**
>
> **[Weakness 1 and Question 1]**
> > **[Weakness 1]** To apply the new analysis in practice, many assumptions have been made, which may not be satisfied when training a real network, so I expect the resulting network obviously outperforms others, so far it's not, perhaps a tighter bound is needed. **[Question 1]** ... the evaluation does not show the proposed method obviously outperforms the composition theorem ...
>
> Thank you for the comments and we hope to clear up these concerns point by point as follows.
>
> Firstly, our hidden-state privacy bound is **significantly tighter** than the composition-based privacy bound (for DP-SGD) after $O(\frac{1}{\lambda\eta} + \frac{n}{b})$, for strongly convex smooth optimization problems. We demonstrate these quantitative comparisons in Sections 4 and 5.
>
> Secondly, in our empirical evaluations of privacy-utility trade-offs (Section 6), the **utility gain** from using our privacy analysis **is significant**. For regularized logistic regression model trained from (handcrafted) ScatterNet features, our hidden state privacy bound enables around 3% higher test accuracy on CIFAR-10, than the baseline of composition-based privacy bound for DP-SGD [36]. Moreover, this accuracy is only 1~2% lower than the non-privately trained logistic regression model on the same features. Please refer to our response to [Weakness 2] for more detailed explanations of these improvements.
>
> Last but not least, we want to highlight that our privacy bound applies to several useful settings, as demonstrated in our experiments in Section 6. This includes training regularized logistic regression models and differentially private fine-tuning. However, we also acknowledge that our current analysis requires several strong conditions to holds, such as strong convexity of the loss functions. Relaxing these conditions in hidden-state privacy analysis is a very important open problem that requires more future work.
>
>
> **[Weakness 2]**
>
> > I am a little confused about the evaluation of the proposed privacy analysis. I see the baseline performance given in this work is lower than in its original work. For example in the experiment on CIFAR10, ScatterNet can actually obtain an accuracy of 69.3%, ResNeXt-29 can achieve 80.0%, SIMCLR can achieve 92.7%, all are on par with using the method proposed in this work.
>
> Thank you for sharing this confusion and hopefully we could clarify as follows.
>
> For experiments in this paper, the baseline we are comparing to is the "ScatterNet + linear" setting in [36], which achieves the best-known accuracy for **linear models** trained from scratch. The baseline test accuracy 66.78$\pm$0.27%$\approx$ 67\% reported in our Table 1 is **the same as** the results for "ScatterNet + linear" setting in [36, Table 1]. Compared to this baseline, we train a logistic regression model that achieves a significantly higher 69.51$\pm$0.23% test accuracy. Moreover, this 69.51$\pm$0.23% accuracy is very close to the optimal non-privately trained logistic regression model (on ScatterNet features), which achieves 71.1% test accuracy. This closeness thus shows the empirical tightness of our privacy dynamics bound for **strongly convex and smooth** optimization problems.
>
> What you are referring to (with "69.3%" accuracy) is a different setting of "ScatterNet + CNN" in [36] that trains deep CNN models on ScatterNet features. This "ScatterNet + CNN" setting is a **non-convex** optimization problem that our current hidden-state privacy bound is **inapplicable** to, therefore is not a baseline setting for comparing our privacy analysis with composition-based bound. However, we still want to highlight that, even when compared to the 69.3% test accuracy of "ScatterNet + CNN" [36], our test accuracy for linear model 69.51$\pm$0.23% ishigher! This high performance of private linear models (on par with or even better than private deep models) is unprecedented in prior literature [36], which again shows that our hidden-state privacy bound is significantly tighter than composition-based bounds for strongly convex and smooth optimization problems.

---

> ### Author Response · Authors · 2022-08-10
> **Follow-up after response**
>
> Thanks again for your time and comments. We just want to reach out to see if our response addresses your main concerns. We are also happy to discuss any further questions or comments that you may have after our response.

---

### Official Review · Reviewer_q4dU · 2022-07-11

**Rating:** 8
**Confidence:** 3
**Soundness:** 4 excellent
**Presentation:** 4 excellent
**Contribution:** 4 excellent

**Summary:**

The paper focuses on improving the privacy analysis of DP-SGD. The paper studies the privacy guarantees when only the last iterate is released. The paper's main contribution in this particular setting is to provide tighter privacy bound for strongly convex loss functions. Finally, the paper provides an example of a classification task where a tighter privacy analysis leads to improved accuracy compared to when composition-based privacy analysis is used.

Overall, I think this is an extremely important result and strongly recommend the paper be accepted.

**Questions:**

1) Could authors comment on the privacy-utility tradeoff when the average of iterates is used instead of the final iterate as the output of the SGD algorithm?


**Limitations:**

Yes.

**Strengths And Weaknesses:**

1) The main result has significant practical implications on differentially private machine learning training.
2) There are many similarities with [14]. Perhaps a minor weakness of the paper is that it isn't entirely clear what are the technical challenges the authors had to overcome to complete the analysis, which were not already addressed in [14].

---

> ### Author Response · Authors · 2022-08-02
> **Response to reviewer q4dU**
>
> **[Weakness]**
> > There are many similarities with [14]. Perhaps a minor weakness of the paper is that it isn't entirely clear what are the technical challenges the authors had to overcome to complete the analysis, which were not already addressed in [14].
>
> We acknowledge that we share some technical similarities with [14]. However, our paper focus on noisy SGD while [14] only studies noisy GD. Consequently, as discussed in Section 3, we incur many new technical difficulties in analyzing the privacy amplifications (unique due to stochastic mini-batches), such as under randomized post-processing and mixture distribution. As we show in Section 3 and Figure 1, *without taking these new privacy amplifications into account*, a direct extension of noisy GD analysis[14] to the mini-batch setting gives a significantly worse privacy bound than ours.
>
> Moreover, to highlight the benefit of our new methodology, we've updated appendix E.7 to show that: **even for noisy GD**, our analysis enables a **tighter privacy bound** than [14] with a conceptually more straightforward proof.
> - We break down one noisy GD update $\theta_{k+1}^0 = \theta_k^0 - \eta \cdot g(\theta_k^0;D) + \sqrt{2\eta\sigma^2}\cdot \mathcal{N}(0,\mathbb{I}_d)$ where $g(\theta_k^j;D) = \frac{1}{n}\sum\_{i=1}^n\nabla\ell(\theta_k^j;x_i)$ into two consecutive steps: a noisy GD update with smaller noise scale followed by pure additive Gaussian noise.
>  1. $\theta_{k}^{\frac{1}{2}} = \theta_k^0 - \eta \cdot g(\theta_k^0;D) + \sqrt{\eta\sigma^2}\cdot \mathcal{N}(0,\mathbb{I}_d)$
>  2. $\theta_{k+1}^0 = \theta_k^{\frac{1}{2}} + \sqrt{\eta\sigma^2}\cdot \mathcal{N}(0,\mathbb{I}_d)$
> - We then view the second step (pure Gaussian noise) as randomized post-processing and apply Lemma 4.2 to quantify the privacy amplification. Meanwhile, for the first step, we view it as Gaussian Mechanism and directly use R\'enyi DP composition.
>
> Following this approach, we derive new privacy bound for noisy GD, which is strictly tighter than the bound in [14]. The complete proof and quantitative comparison are in Appendix E.7. This shows the advantage of our new analysis methodology compared to [14].
>
> **[Question]**
>
> > Could authors comment on the privacy-utility trade-off when the average of iterates is used instead of the final iterate as the output of the SGD algorithm?
>
>
> Thanks for this question. We do not have a definite answer, but we are happy to share our thoughts.
>
> - Theoretically speaking, to the best of our knowledge, the utility bound for last-iterate is typically worse than that of average-iterate. This is because last-iterate utility bounds are more challenging to prove and tend to be quantitatively worse than average-iterate. Therefore, for utility analysis, standard results[a] usually analyze the excess empirical/population risk for the average of iterates.
> - However, privacy bound for the last iterate tend to be better (smaller) than that for the average-iterate, because of the additional hidden-state privacy amplification. In certain scenarios, these hidden-state amplifications enable privacy bounds for the last iterate (e.g. what we are studying in this paper) to converge to a constant, regardless of how many iterations there are in the training. On the contrary, to the best of our knowledge, no such hidden-state privacy amplification are known for the average of iterates. Consequently, existing privacy analysis for average-iterate boils down to R\'enyi DP composition over the iterates, and worsens with the number of iterates.
> - In terms of theoretical privacy utility trade-off, a large body of literature, e.g., [a,b], proves that the average of iterates achieves optimal privacy-utility trade-off in certain regimes. However, for convex stochastic optimization, by exploiting the privacy amplification of last-iterate, recent works[c] proved that last-iterate also achieves an asymptotically optimal privacy-utility trade-off (for an appropriate choice of step sizes). Therefore, we believe it is still an interesting open problem as to whether average-iterate or last-iterate enables a better privacy utility trade-off for more optimization regimes.
> - In terms of trade-off between empirical utility and theoretical privacy bound, to the best of our knowledge, most SOTA trade-offs are achieved via the last-iterate model (instead of average of iterates). This is also one of the most important motivations for focusing on privacy bound for the last iterate in this paper.
>
> **References**
>
> [a] Bassily, R., Smith, A., Thakurta, A. Private empirical risk minimization: Efficient algorithms and tight error bounds. FOCS, 2014.
>
> [b] Bassily, R., Feldman, V., Talwar, K., and Guha Thakurta, A. Private stochastic convex optimization with optimal rates. NIPS, 2019.
>
> [c] Feldman, V., Koren, T., Talwar, K. Private stochastic convex optimization: optimal rates in linear time. STOC, 2020.

---

### Official Review · Reviewer_duNW · 2022-07-11

**Rating:** 7
**Confidence:** 4
**Soundness:** 4 excellent
**Presentation:** 1 poor
**Contribution:** 3 good

**Summary:**

This paper extends the last-iterate analysis to DP-SGD on strongly convex and smooth loss functions. The main contribution, compared the similar analysis GD, is to take two kinds of amplification into account: 1) post-processing by adding Gaussian noise; (2) sub-sampling. The empirical results show that the proposed methods outperforms composition-based algorithms.

**Questions:**

1. Seems the only reason to assume strong convexity is for the Lipchitzness and utility result? Can we just assume Lipchitzness for the privacy guarantee? Please correct me if I miss anything.
2. How does the results in the paper compare to a parallel work "Langevin Diffusion: An Almost Universal Algorithm for Private Euclidean (Convex) Optimization"?

**Limitations:**

1. The writing is not good enough.
2. Many details of the evaluation is missing.

**Strengths And Weaknesses:**

This paper makes a solid step towards last-iterate analysis for DP-SGD. Although there is some space for improvement, I believe the paper surely passes the bar for acceptance.

Strength:
1. Last-iterate analysis is a very important direction to further improve privacy-utility trade-off for DP-SGD. The authors manage to augment the previous analysis for GD with amplification by post-processing and sub-sampling, which is a solid contribution.

Weakness:
1. The paper is not well written enough. There is huge space for improvement on the organization of the sections, the section names and the way to present the contributions. For example, even with some background knowledge, the titles for section 4 and 5 are somewhat confusing at the first glance for me. I suggest merging 4 and 5 into one section because they are actually two steps in one proof. There are also some sentences that do not read well (e.g. Line 109). Uppercase and lowercase letters are not used in a universal way in section titles. Some citations are outdated. For example, Tramer and Boneh 2022 are accepted to ICLR 2022 but the cited one is still the arXiv version.
2. Table 2 does not report what kind of clipping is used? Also the number of iterations used in the experiments are not reported.
3. Although hyperparameter tuning is typically not taken into consideration when counting privacy budget, the authors should still expose the details of their hyper-parameter tuning process.

---

> ### Author Response · Authors · 2022-08-02
> **Response to Reviewer duNW**
>
>
>  **[Weakness 1]**
>  > ... huge space for improvement on the organization ... the section names and the way to present ... some citations are outdated...
>
>  Thank you for the comments. We have updated the section names and the outdated citations in the revised paper.
>
> **[Weakness 2&3]**
>  > Table 2 does not report what kind of clipping is used? Also the number of iterations used in the experiments are not reported ... the authors should still expose the details of their hyper-parameter tuning process.
>
>  **The clipping method** Table 2 uses the same clipping method as Table 1, i.e., clipping feature and gradient under privacy dynamics analysis, while only clipping gradient under DP-SGD analysis. We have updated Table 2 to clarify the clipping method.
>
>  **Number of iterations used:** The optimal accuracy under privacy dynamics analysis is achieved after over 1200 epochs for training from scratch (600 epochs for fine-tuning). Meanwhile, DP-SGD achieves the optimal accuracy at lower than 60 epochs for training from scratch (40 epochs for fine-tuning). We believe that the ability to tolerate a longer training process (under a constrained privacy budget) is the essential factor that enables better accuracy under our hidden-state privacy analysis than composition-based analysis.
>
>  **Details regarding hyper-parameter tuning:** Due to limited space, we had to put details regarding hyper-parameter tuning in Appendix E.6-E.8. Both our hyperparameter tuning process and code are directly adapted from Tramèr and Boneh, 2021. We use grid search, and an example hyperparameter tuning script for cifar10 is in `run_baselines_cifar10_feature_clip.py`.
>
>
> **[Question 1]**
> > Seems the only reason to assume strong convexity is for the Lipchitzness and utility result? Can we just assume Lipchitzness for the privacy guarantee?
>
> Thank you for this question. It is indeed an important open problem to relax the strong convexity condition in our hidden-state privacy bounds. However, in our current analysis, strong convexity **is necessary** for obtaining a **converging** hidden-state privacy bound. This is for ensuring that log-Sobolev inequality with constant $c$ (which is a condition required by Lemma 4.1) holds throughout the training process (i.e. after any number of epochs $k$ and iterations $j$). **Without strong convexity**, the LSI constant $c$ may worsen (becomes smaller) as the training process continues. E.g. the LSI constant proved in Lemma C.2 decreases to zero as the number of epochs $k$ and iterations $j$ grow. Consequently, under this small $c\rightarrow0$, the amplification ratio $(1 + \frac{c\cdot 2\eta \sigma^2}{L^2})$ in Lemma 4.1 becomes close to 1, and is not small enough to amplify the additive increase in privacy loss in preceding steps, thus failing to enable a converging privacy loss bound.
>
> **[Question 2]**
> > How does the results in the paper compare to a parallel work "Langevin Diffusion: An Almost Universal Algorithm for Private Euclidean (Convex) Optimization"?
>
> Thank you for sharing this interesting work. We were unaware of this work (abbreviated as [GTU22]) at submission time, as it was released on arXiv around one month before the deadline. But we are happy to add a discussion in future revisions. What we could observe at this stage are several qualitative differences between our paper and [GTU22].
>  - Our analysis holds for a concrete discretized noisy SGD algorithm. Meanwhile, [GTU22] focus on a conceptual continuous-time algorithm (Langevin Diffusion). As an analogy, this algorithmic difference between [GTU22] and us is similar to how continuous gradient flow differs from GD/SGD.
>  - Our analysis allows a more general form of initialization distribution. More specifically, our privacy bounds holds under arbitrary initialization distribution in Algorithm 1. On the contrary, the results in [GTU22, Section 7.1] require that the initialization distribution is Gaussian.
>  - Our analysis does not require convergence of the algorithm parameters $\theta_K^0$ to a stationary distribution as $K\rightarrow\infty$. E.g., even if the loss function changes its form in every iteration, our privacy bound still holds (as long as all the used loss functions are strongly convex and smooth). On the contrary, the converging privacy bound in [GTU22, Section 7.1] heavily relies on 1) the convergence of the Langevin diffusion process to a stationary Gibbs distribution and 2) the closeness between stationary distributions when running the algorithm on neighboring datasets.
>
> > Many details of the evaluation is missing.
>
> Due to limited space and our paper's focus on proving tighter theoretical privacy bounds, we had to put most experiment details in Appendix E. Please also refer to our attached code for more information regarding the experiments.

---

### Meta-Review · Area_Chair_xiE1 · 2022-08-25

**Recommendation:** Accept
**Confidence:** Certain

**Metareview:**

This work shows that for strongly convex and smooth loss functions, running DPSGD has a privacy cost that stops growing at some point. The answers a question that has been open and is of significant theoretical and practical interest. They show empirically that this new result can allows one to get better privacy-accuracy trade-offs in some cases.
This work is a big step ahead in analysis of DPSGD and I recommend acceptance.

**Award:**

No

---

### Decision · Program_Chairs · 2022-09-14

Accept